# SEMANTIC-AWARE WASSERSTEIN POLICY REGULARIZATION FOR LARGE LANGUAGE MODEL ALIGNMENT

**Byeonghu Na[1], Hyungho Na[2], Yeongmin Kim[1], Suhyeon Jo[1], HeeSun Bae[1], Mina Kang[1], Il-Chul Moon[1,3]**
[1]KAIST, [2]UNIST, [3]summary.ai
byeonghu.na@kaist.ac.kr, h.na@unist.ac.kr
{alsdudrla10,suhyeonjo,cat2507,kasong13,icmoon}@kaist.ac.kr

## ABSTRACT

Large language models (LLMs) are commonly aligned with human preferences using reinforcement learning from human feedback (RLHF). In this method, LLM policies are generally optimized through reward maximization with Kullback-Leibler (KL) divergence regularization of the reference policy. However, KL and its $f$-divergence variants only compare token probabilities at identical indices, failing to capture semantic similarity. We propose Wasserstein Policy Regularization (WPR), a semantic-aware regularization for the RLHF framework based on the entropy-regularized Wasserstein distance, which incorporates the geometry of the token space. The dual formulation of the distance expresses the regularization as penalty terms applied to the reward via optimal dual variables, which yield a tractable objective compatible with standard RL algorithms. Empirically, our method outperforms KL- and $f$-divergence-based baselines, demonstrating the benefits of semantic-aware policy distances for alignment. Our code is available at https://github.com/aailab-kaist/WPR.

## 1 INTRODUCTION

Large language models (LLMs) have achieved remarkable progress in recent years, powering applications ranging from conversational agents to code generation (Touvron et al., 2023; Achiam et al., 2023; Hui et al., 2024). A central challenge in their deployment is aligning model behavior with human preferences. Reinforcement learning from human feedback (RLHF) has emerged as the dominant paradigm for alignment, where models are optimized to better reflect user intent (Christiano et al., 2017; Bai et al., 2022; Ouyang et al., 2022). The standard RLHF pipeline trains a reward model from human preference data and optimizes the LLM policy to maximize reward while remaining close to a supervised fine-tuned reference model (Ouyang et al., 2022). Recent advances such as Direct Preference Optimization (DPO) (Rafailov et al., 2023) and its variants (Azar et al., 2024; Ethayarajh et al., 2024) follow a similar principle, reducing the preference learning to implicit reward maximization with reverse Kullback–Leibler (KL) regularization to maintain the reference policy.

The policy regularization by the KL divergence is widely adopted because the KL divergence can be computed directly from the token probabilities of the reference and the trained models, which is implemented as a penalty on the reward. While KL-based regularization is effective in practice, it exhibits known shortcomings; for example, the reverse KL tends to be mode-seeking, which reduces output diversity. Recent works have addressed these issues by replacing reverse KL with alternative $f$-divergences, such as $f$-DPO (Wang et al., 2024) and $\chi$PO (Huang et al., 2025). However, these $f$-divergence-based constraints still measure policy discrepancy only by comparing token probabilities at identical indices, thereby ignoring semantic relationships between tokens.

To illustrate this limitation, we introduce a simple example in Figure 1. We consider a vocabulary {cat, kitten, dog, table} and compare a reference policy $\pi_{\text{ref}}$ and two learned policies, $\pi_1$ and $\pi_2$, in the context of next token selection when answering the question "*What is in this image?*" given a small cat image. In this example, $\pi_{\text{ref}}$, $\pi_1$, and $\pi_2$ assign high probability mass to cat, kitten, and table, respectively. Semantically, (cat, kitten) is more closely related than (cat, table), so we would expect $\pi_{\text{ref}}$ to be closer to $\pi_1$ than $\pi_2$. However, KL values diverge

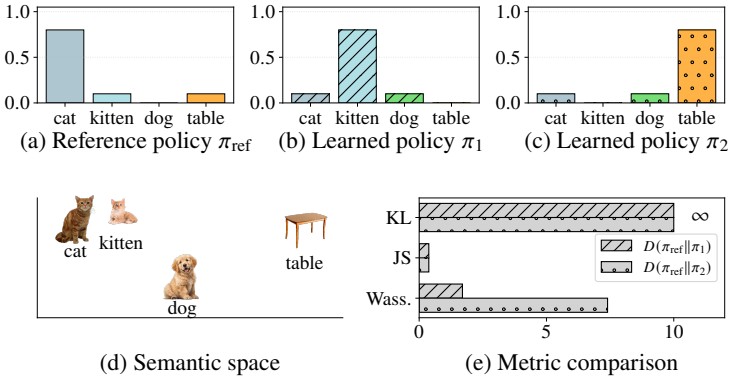

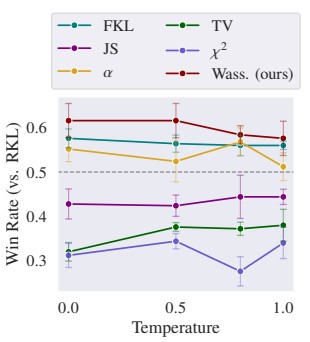

Figure 2: Win rates against KL-based regularization across sampling temperatures on dialogue generation with Gemma-2B, comparing $f$-divergences and our Wasserstein distance.

Figure 1: Motivating example for the Wasserstein distance in LLM policy comparison. (a-c) Probability distributions of the reference and learned policies. (d) Semantic space among tokens. (e) Comparison under different divergences, where Wasserstein distance captures semantic relationships that KL and JS divergences fail to reflect.

due to the support mismatch, and other $f$-divergences such as Jensen-Shannon (JS) divergence assign the same distance to $\pi_1$ and $\pi_2$, failing to reflect semantic proximity.

To overcome this limitation, we introduce a new RLHF regularization framework based on Wasserstein distances, which we refer to as Wasserstein Policy Regularization (WPR). Unlike the KL and other $f$-divergences, the Wasserstein metric compares distributions by explicitly considering the geometry of the underlying token space. This enables flexible, user-defined cost functions that naturally encode semantic similarity between tokens. Additionally, it remains well-defined even when the support of two distributions does not overlap. In the context of language modeling, these properties are crucial because policies that assign high probability to semantically related tokens (e.g., cat and kitten) could likewise be regarded as similar. As illustrated in Figure 1, the Wasserstein distance properly identifies the reference policy $\pi_{\text{ref}}$ as being closer to $\pi_1$ than to $\pi_2$, thereby capturing semantic proximity that KL and other $f$-divergences fail to reflect. As a result, as shown in Figure 2, the policy regularization with the Wasserstein distance achieves superior generation performance compared to KL and other $f$-divergence-based approaches, with experimental details provided in Section 5.2.

Building on these properties, we propose a tractable optimization framework that leverages the entropy-regularized Wasserstein distance, i.e., Sinkhorn distance (Cuturi, 2013), as a semantic-aware policy regularizer. Computing this distance requires solving an entropic optimal transport problem; we recast it in the dual and show that the resulting dual variables represent the regularization penalty. This penalty can be incorporated into the reward as token-wise adjustments, analogous to standard KL-based regularization, making the formulation compatible with standard RL algorithms such as PPO (Schulman et al., 2017). The optimal dual variables can be obtained efficiently via the Sinkhorn algorithm with modest overhead. Empirically, our approach outperforms KL- and $f$-divergence–based baselines, highlighting the effectiveness of semantic-aware policy distances for RLHF.

## 2 RELATED WORKS

**Aligning Large Language Models** Traditional supervised fine-tuning (SFT) methods have been effective in language generation but shows limitations in aligning outputs with human preferences, such as sentiment (Maas et al., 2011), helpfulness (Askell et al., 2021), harmlessness (Gehman et al., 2020), and truthfulness (Lin et al., 2022). RLHF has become the standard approach for preference alignment (Stiennon et al., 2020; Ouyang et al., 2022). It trains a reward model from human preference data and uses it to optimize the policy via reinforcement learning to better match human preferences (Christiano et al., 2017; Ziegler et al., 2019; Böhm et al., 2019). This approach has enabled successful LLMs such as ChatGPT (Achiam et al., 2023). Recent alternatives avoid explicit reward models, including RAFT (Dong et al., 2023), RRHF (Yuan et al., 2023), and DPO (Rafailov et al., 2023), which reformulate preference alignment as direct policy optimization.

**Regularization for Policy Learning**    Methods such as RLHF and DPO incorporate regularization by a reverse KL divergence during preference alignment to prevent the learned policy from deviating significantly from a reference model trained via SFT. While this constrains learning to remain close to the behavior of the reference model, the mode-seeking nature of reverse KL tends to limit output diversity (Wiher et al., 2022; Khalifa et al., 2021; Perez et al., 2022; Glaese et al., 2022). To address this limitation, studies such as $f$-DPO (Wang et al., 2024) and $\chi$PO (Huang et al., 2025) have been proposed. In parallel, other works (Han et al., 2025; Kim et al., 2025) explore alternative divergences for directly matching the optimal policy, though our focus in this work is on regularization. However, $f$-divergence-based methods share a key limitation: they measure the distributional discrepancy solely based on probability values at identical indices, without reflecting the semantic relationships between tokens. In contrast, we propose a novel approach that leverages distance metrics from the Integral Probability Metric (IPM) (Müller, 1997), such as Wasserstein distance (Adler & Lunz, 2018; Panaretos & Zemel, 2019), to enable semantic-aware policy regularization.

**Application of Wasserstein Distance**    The Wasserstein distance and its variants, such as the Sinkhorn distance, have been widely applied across many machine learning domains, including generative modeling, robust optimization, and reinforcement learning (Arjovsky et al., 2017; Sinha et al., 2018; Moskovitz et al., 2021; Song et al., 2023; Cui et al., 2024). For example, in generative modeling, Wasserstein GANs (Arjovsky et al., 2017) leverage the Wasserstein distance between the generator distribution and the data distribution to improve training stability and mitigate mode collapse. In robust optimization, adversarial training is formulated using Wasserstein balls around the data distribution to provide certified robustness (Sinha et al., 2018). In reinforcement learning, the Wasserstein natural gradient aligns policy updates with the local optimal-transport geometry in behavioral policy optimization (Moskovitz et al., 2021). Song et al. (2023) explore trust-region policy optimization based on Wasserstein and Sinkhorn distance. Building on this line of work, we explore Wasserstein regularization for RLHF, enabling semantic-aware policy alignment.

## 3  PRELIMINARY

### 3.1  WASSERSTEIN DISTANCE

The Wasserstein distance between two distributions $\pi$ and $\pi'$ is defined as

$$D_{\mathrm{W}}(\pi||\pi') := \min_{\boldsymbol{P} \in U(\pi,\pi')} \mathbb{E}_{(y,y') \sim \boldsymbol{P}}\big[c(y,y')\big] = \min_{\boldsymbol{P} \in U(\pi,\pi')} \langle \boldsymbol{P}, \boldsymbol{C} \rangle, \tag{1}$$

where $U(\pi,\pi') := \{\boldsymbol{P} \in \mathbb{R}_+^{d \times d} | \boldsymbol{P}\mathbf{1}_d = \pi, \boldsymbol{P}^\top \mathbf{1}_d = \pi'\}$ is the set of couplings between $\pi$ and $\pi'$, $\boldsymbol{C} \in \mathbb{R}_+^{d \times d}$ is the cost matrix with entries $\boldsymbol{C}_{y,y'} := c(y,y') \geq 0$, $\langle \cdot, \cdot \rangle$ denotes the Frobenius inner product, and $d$ is the cardinality of the outcome space.

To obtain a smooth and computationally tractable approximation to the Wasserstein distance, an entropy regularization term is added to the optimal transport objective, yielding the entropy-regularized Wasserstein distance, also known as the Sinkhorn distance (Cuturi, 2013):

$$D_{\tilde{\mathrm{W}}}(\pi||\pi') := \min_{\boldsymbol{P} \in U(\pi,\pi')} \left\{ \langle \boldsymbol{P}, \boldsymbol{C} \rangle - \frac{1}{\lambda}\mathcal{H}(\boldsymbol{P}) \right\}, \tag{2}$$

where $\lambda$ is an entropy regularization hyperparameter, and $\mathcal{H}(\boldsymbol{P}) := - \sum_{i=1}^d \sum_{j=1}^d P_{ij}(\log P_{ij} - 1)$ is the entropy regularization term, equivalent to the Shannon entropy up to an additive constant.

While the Wasserstein distance directly relies on the Kantorovich dual formulation of optimal transport, the Sinkhorn distance arises from the dual of its entropically regularized variant (Villani, 2008; Peyré et al., 2019):

$$D_{\mathrm{W}}(\pi||\pi') = \max_{\boldsymbol{\phi},\boldsymbol{\psi}} \left\{ \sum_{i=1}^d \phi_i \pi_i + \sum_{j=1}^d \psi_j \pi'_j \ \bigg| \ \phi_i + \psi_j \leq C_{ij} \ \forall i,j \right\}, \tag{3}$$

$$D_{\tilde{\mathrm{W}}}(\pi||\pi') = \max_{\boldsymbol{\phi},\boldsymbol{\psi}} \left\{ \sum_{i=1}^d \phi_i \pi_i + \sum_{j=1}^d \psi_j \pi'_j - \frac{1}{\lambda} \sum_{i=1}^d \sum_{j=1}^d \exp\left(\lambda(\phi_i + \psi_j - C_{ij})\right) \right\}, \tag{4}$$

where $\phi$ and $\psi$ are the dual variables. In the Wasserstein case with the Euclidean cost, the dual variables reduce to a single 1-Lipschitz function, which is typically parameterized by a critic network and optimized with gradient-based methods (Arjovsky et al., 2017). In contrast, the entropy-regularized formulation yields dual optimality conditions corresponding to matrix scaling factors, which can be computed efficiently by the Sinkhorn-Knopp algorithm (Sinkhorn & Knopp, 1967) as closed-form iterations alternating between row and column normalization (Cuturi, 2013; Cuturi & Doucet, 2014).

The entropy-regularized Wasserstein distance produces smoother and denser couplings between distributions, and it converges to the Wasserstein distance as $\lambda \to \infty$. Moreover, compared to the unregularized Wasserstein distance, the Sinkhorn distance can be computed more efficiently, incurring substantially less computational overhead. Since our setting requires computing next-token predictive distributions conditioned on various prompts and partial responses, we employ the entropic regularization variant rather than the critic-based Wasserstein distance, as the former admits the closed-form iterations.[1]

## 3.2 REINFORCEMENT LEARNING FROM HUMAN PREFERENCES (RLHF)

Our goal is to align an autoregressive LLM, denoted as $\pi_{\boldsymbol{\theta}}(\mathbf{y}|\mathbf{x})$ where $\mathbf{x}$ is a user prompt and $\mathbf{y}$ is a response, with human preferences through reinforcement learning (RL). The RLHF procedure consists of three main stages. First, we perform supervised fine-tuning (SFT) to obtain a reference model $\pi_{\text{ref}}$, which serves as the initial aligned model. Second, we train a reward model $r(\mathbf{x}, \mathbf{y})$ on a preference dataset, enabling the estimation of scalar rewards for responses $\mathbf{y}$ given prompts $\mathbf{x}$. Finally, using both the reference model $\pi_{\text{ref}}$ and the reward model $r$, we optimize the following objective to fine-tune the language model $\pi_{\boldsymbol{\theta}}$:

$$\max_{\pi_{\boldsymbol{\theta}}} \mathcal{J}(\pi_{\boldsymbol{\theta}}; \pi_{\text{ref}}) := \mathbb{E}_{\mathbf{x} \sim \mathcal{D}} \left[ \mathbb{E}_{\mathbf{y} \sim \pi_{\boldsymbol{\theta}}(\mathbf{y}|\mathbf{x})} \left[ r(\mathbf{x}, \mathbf{y}) \right] - \beta D \left( \pi_{\boldsymbol{\theta}}(\mathbf{y}|\mathbf{x}) || \pi_{\text{ref}}(\mathbf{y}|\mathbf{x}) \right) \right], \tag{5}$$

where $D$ denotes a policy divergence, $\beta$ is a policy regularization hyperparameter, and $\mathcal{D}$ is the prompt dataset. This objective encourages the policy to generate responses that maximize reward while remaining close to the reference model.

In most of the previous works, the divergence $D$ is instantiated as the (reverse) KL divergence:

$$\max_{\pi_{\boldsymbol{\theta}}} \mathcal{J}_{\text{KL}}(\pi_{\boldsymbol{\theta}}; \pi_{\text{ref}}) := \mathbb{E}_{\mathbf{x} \sim \mathcal{D}} \left[ \mathbb{E}_{\mathbf{y} \sim \pi_{\boldsymbol{\theta}}(\mathbf{y}|\mathbf{x})} \left[ r(\mathbf{x}, \mathbf{y}) \right] - \beta D_{\text{KL}} \left( \pi_{\boldsymbol{\theta}}(\mathbf{y}|\mathbf{x}) || \pi_{\text{ref}}(\mathbf{y}|\mathbf{x}) \right) \right] \tag{6}$$

$$= \mathbb{E}_{\mathbf{x}} \left[ \sum_{n=1}^{N} \mathbb{E}_{y_n \sim \pi_{\boldsymbol{\theta}}(y_n|\mathbf{x}, \mathbf{y}_{1:n-1})} \left[ R(\mathbf{x}, \mathbf{y}_{1:n}) \right] - \beta \sum_{n=1}^{N} D_{\text{KL}} \left( \pi_{\boldsymbol{\theta}}(y_n|\mathbf{x}, \mathbf{y}_{1:n-1}) || \pi_{\text{ref}}(y_n|\mathbf{x}, \mathbf{y}_{1:n-1}) \right) \right] \tag{7}$$

$$= \mathbb{E}_{\mathbf{x}} \left[ \sum_{n=1}^{N} \mathbb{E}_{y_n \sim \pi_{\boldsymbol{\theta}}(y_n|\mathbf{x}, \mathbf{y}_{1:n-1})} \left[ R(\mathbf{x}, \mathbf{y}_{1:n}) - \beta \log \frac{\pi_{\boldsymbol{\theta}}(y_n|\mathbf{x}, \mathbf{y}_{1:n-1})}{\pi_{\text{ref}}(y_n|\mathbf{x}, \mathbf{y}_{1:n-1})} \right] \right], \tag{8}$$

where $D_{\text{KL}}(\pi(y_n) || \pi'(y_n)) := \mathbb{E}_{y_n \sim \pi(y_n)} \left[ \log \frac{\pi(y_n)}{\pi'(y_n)} \right]$, $N$ is the sequence length of $\mathbf{y}$, and $R(\mathbf{x}, \mathbf{y}_{1:n}) = r(\mathbf{x}, \mathbf{y}_{1:N})$ for $n = N$, and 0 otherwise. As shown in Eq. (8), the KL regularization term $D_{\text{KL}}$ can be rewritten as the expectation of the log-ratio between the two policies, which allows standard RL algorithms such as PPO to be applied for optimization. Furthermore, several works (Wang et al., 2024; Huang et al., 2025) have generalized the KL divergence to other $f$-divergences and developed tractable optimization formulations accordingly.

However, as mentioned in the Introduction, KL or other $f$-divergence measures compare policies solely by token-level probability differences at identical indices, without accounting for the underlying semantic structure of tokens. This limitation prevents them from fully capturing meaningful distributional differences in language generation. To address this, we replace the divergence term with the Wasserstein distance, more precisely the Sinkhorn distance, which naturally incorporates semantic information, and we develop a tractable optimization framework for this objective.

---

[1]In preliminary experiments, we explored the critic-based Wasserstein distance but found that the resulting policy regularization was insufficient, leading to suboptimal performance.

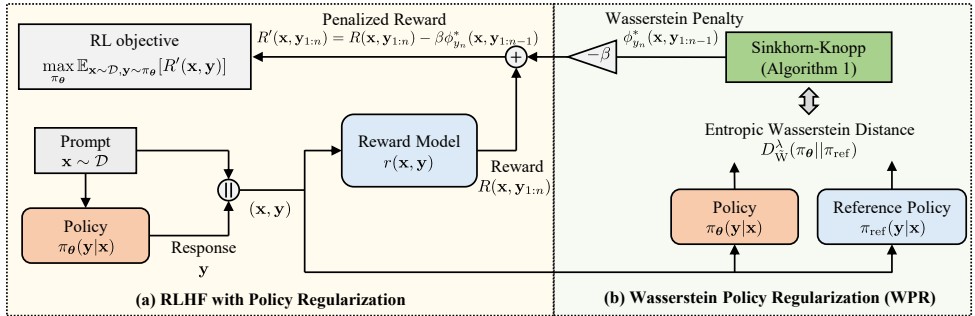

Figure 3: Overview of RLHF with Wasserstein Policy Regularization. (a) Standard RLHF with a policy regularization penalty. (b) Our proposed Wasserstein policy regularization, where the penalty is computed from the optimal dual variables obtained via the Sinkhorn-Knopp algorithm.

## 4 METHOD: WASSERSTEIN POLICY REGULARIZATION

### 4.1 RLHF OBJECTIVE WITH WASSERSTEIN POLICY REGULARIZATION

In this section, we propose Wasserstein Policy Regularization (WPR), which regularizes LLM policies in RLHF using the Wasserstein distance as the statistical distance between policies. We formulate the Wasserstein-regularized RLHF objective by replacing the token-level KL divergence regularization term in Eq. (7) with a Wasserstein regularization term:

$$\max_{\pi_{\boldsymbol{\theta}}} \mathcal{J}_{\mathrm{W}}(\pi_{\boldsymbol{\theta}}; \pi_{\mathrm{ref}}) := \tag{9}$$

$$\mathbb{E}_{\mathbf{x}\sim\mathcal{D}}\left[\sum_{n=1}^{N}\mathbb{E}_{y_n\sim\pi_{\boldsymbol{\theta}}(y_n|\mathbf{x},\mathbf{y}_{1:n-1})}\left[R(\mathbf{x},\mathbf{y}_{1:n})\right] - \beta\sum_{n=1}^{N}D_{\mathrm{W}}\left(\pi_{\boldsymbol{\theta}}(y_n|\mathbf{x},\mathbf{y}_{1:n-1})||\pi_{\mathrm{ref}}(y_n|\mathbf{x},\mathbf{y}_{1:n-1}))\right)\right]$$

Here, $D_{\mathrm{W}}\left(\pi_{\boldsymbol{\theta}}(y_n|\mathbf{x},\mathbf{y}_{1:n-1})||\pi_{\mathrm{ref}}(y_n|\mathbf{x},\mathbf{y}_{1:n-1})\right) := \min_{\boldsymbol{P}^{(n)}\sim U_n(\pi_{\boldsymbol{\theta}},\pi_{\mathrm{ref}})} < \boldsymbol{P}^{(n)}, \boldsymbol{C} >$ where $U_n(\pi_{\boldsymbol{\theta}}, \pi_{\mathrm{ref}}) := \left\{\boldsymbol{P}^{(n)} \in \mathbb{R}_+^{d\times d} | \boldsymbol{P}^{(n)}\mathbf{1}_d = \pi_{\boldsymbol{\theta}}(\cdot|\mathbf{x},\mathbf{y}_{1:n-1}), \boldsymbol{P}^{(n)\top}\mathbf{1}_d = \pi_{\mathrm{ref}}(\cdot|\mathbf{x},\mathbf{y}_{1:n-1})\right\}$, $\boldsymbol{C} \in \mathbb{R}_+^{d\times d}$ is the cost matrix, $d$ is the token dictionary size, and $N$ is the sequence length of $\mathbf{y}$. It should be noted that $\boldsymbol{P}^{(n)}$ depends on $(\mathbf{x}, \mathbf{y}_{1:n-1})$, but we omit these input terms for simplicity.

The next step is to formulate the Wasserstein distance between two token-level discrete distributions so as to obtain a tractable optimization objective for $\pi_{\boldsymbol{\theta}}$. However, computing the exact Wasserstein distance requires solving a linear program, which quickly becomes intractable when the distributional support is large (Kuhn et al., 2019). As discussed in Section 3.1, a widely used approximation is to introduce entropic regularization into the transport problem, referred to as the entropy-regularized Wasserstein distance or Sinkhorn distance (Cuturi, 2013):

$$\max_{\pi_{\boldsymbol{\theta}}} \mathcal{J}_{\tilde{\mathrm{W}}}(\pi_{\boldsymbol{\theta}}; \pi_{\mathrm{ref}}) := \tag{10}$$

$$\mathbb{E}_{\mathbf{x}\sim\mathcal{D}}\left[\sum_{n=1}^{N}\mathbb{E}_{y_n\sim\pi_{\boldsymbol{\theta}}(y_n|\mathbf{x},\mathbf{y}_{1:n-1})}\left[R(\mathbf{x},\mathbf{y}_{1:n})\right] - \beta\sum_{n=1}^{N}D_{\tilde{\mathrm{W}}}^{\lambda}\left(\pi_{\boldsymbol{\theta}}(y_n|\mathbf{x},\mathbf{y}_{1:n-1})||\pi_{\mathrm{ref}}(y_n|\mathbf{x},\mathbf{y}_{1:n-1}))\right)\right]$$

where $D_{\tilde{\mathrm{W}}}^{\lambda}\left(\pi_{\boldsymbol{\theta}}(\cdot|\mathbf{x},\mathbf{y}_{1:n-1})||\pi_{\mathrm{ref}}(\cdot|\mathbf{x},\mathbf{y}_{1:n-1})\right) := \min_{\boldsymbol{P}^{(n)}\in U_n}\left\{\langle\boldsymbol{P}^{(n)},\boldsymbol{C}\rangle - \frac{1}{\lambda}\mathcal{H}(\boldsymbol{P}^{(n)})\right\}. \tag{11}$

We refer to this objective, $\mathcal{J}_{\tilde{\mathrm{W}}}(\pi_{\boldsymbol{\theta}}; \pi_{\mathrm{ref}})$, as the entropic Wasserstein-regularized RLHF objective. We now derive the dual problem from the regularized primal transportation problem in Eq. (11). Specifically, we introduce the Lagrangian function $\mathcal{L}$ corresponding to Eq. (11).

$$\mathcal{L}(\boldsymbol{P}^{(n)}, \boldsymbol{\phi}, \boldsymbol{\psi}) := \sum_{i=1}^{d}\sum_{j=1}^{d}\left(P_{ij}^{(n)}C_{ij} + \frac{1}{\lambda}P_{ij}^{(n)}(\log P_{ij}^{(n)} - 1)\right)$$

$$+ \sum_{i=1}^{d}\phi_i([\pi_{\boldsymbol{\theta}}]_i - \sum_{k=1}^{d}P_{ik}^{(n)}) + \sum_{j=1}^{d}\psi_j([\pi_{\mathrm{ref}}]_j - \sum_{k=1}^{d}P_{kj}^{(n)}), \tag{12}$$

where $\{\phi_i\}_{i=1}^d$ and $\{\psi_j\}_{j=1}^d$ are the Lagrange multipliers, introduced to enforce the marginal constraints in $U_n$; specifically, they ensure that the row sums of $\boldsymbol{P}^{(n)}$ match $\pi_{\boldsymbol{\theta}}(\cdot \mid \mathbf{x}, \mathbf{y}_{1:n-1})$ and the column sums match $\pi_{\text{ref}}(\cdot \mid \mathbf{x}, \mathbf{y}_{1:n-1})$. Similar to $\boldsymbol{P}^{(n)}$, the dual variables $\phi$ and $\psi$ are functions of $(\mathbf{x}, \mathbf{y}_{1:n-1})$, but we omit their input terms for brevity unless this causes ambiguity. Based on this Lagrangian, the corresponding dual problem is given by

$$\max_{\phi,\psi} \sum_{i=1}^{d} \phi_i [\pi_{\boldsymbol{\theta}}]_i + \sum_{j=1}^{d} \psi_j [\pi_{\text{ref}}]_j - \sum_{i=1}^{d} \sum_{j=1}^{d} \frac{1}{\lambda} \exp(\lambda(\phi_i + \psi_j - C_{ij})), \quad (13)$$

which is derived in Appendix A.1. With strong duality and formulation of the primal solution, we can find the optimal solutions by Proposition 1 (Cuturi & Doucet, 2014).

**Proposition 1.** *(Cuturi & Doucet, 2014) There exists a pair of vectors $(\mathbf{u}, \mathbf{v}) \in \mathbb{R}_+^d \times \mathbb{R}_+^d$ such that the optimal solutions of $\boldsymbol{P}^{(n)}$, $\phi$, and $\psi$ are respectively given by*

$$\boldsymbol{P}^{(n)^*} = \text{diag}(\mathbf{u}) \exp(-\lambda \boldsymbol{C}) \text{diag}(\mathbf{v}), \quad \phi^* = -\frac{1}{\lambda} \log(\mathbf{u}), \quad \psi^* = -\frac{1}{\lambda} \log(\mathbf{v}). \quad (14)$$

We present the proof in Appendix A.2. Note that in our formulation, $\exp$ denotes the element-wise exponential applied to each entry of $\boldsymbol{C}$. In addition, for any real value $t$, the pair of dual variables, $(\phi + t\mathbf{1}_d, \psi - t\mathbf{1}_d)$ yields the same dual objective value. Hence, the dual optimal solutions are not unique but are determined only up to an additive constant. However, as shown in Theorem 2, when formulating the policy optimization problem based on this optimal solution, the additive term remains constant with respect to the policy and can therefore be ignored, yielding an equivalent problem.

By strong duality, substituting the optimal primal and dual variables obtained in Proposition 1 into the objective in Eq. (13) yields an expression of the entropy-regularized Wasserstein distance in terms of the optimal variables. Plugging this result back into the RLHF formulation in Eq. (10), we obtain a tractable optimization problem, as stated in Theorem 2.

**Theorem 2.** *Let $\phi^*(\mathbf{x}, \mathbf{y}_{1:n-1})$ denote the optimal dual variables of the entropic optimal transport problem in Eq. (13). Then, the entropic Wasserstein-regularized RLHF in Eq. (10) can be equivalently written as a reward maximization problem with an additional penalty, induced by $\phi^*$, i.e.,*

$$\mathcal{J}_{\tilde{W}}(\pi_{\boldsymbol{\theta}}; \pi_{ref}) = \mathbb{E}_{\mathbf{x}\sim\mathcal{D}} \left[ \sum_{n=1}^{N} \mathbb{E}_{y_n \sim \pi_{\boldsymbol{\theta}}(y_n|\mathbf{x},\mathbf{y}_{1:n-1})} \left[ R(\mathbf{x}, \mathbf{y}_{1:n}) - \beta \phi_{y_n}^*(\mathbf{x}, \mathbf{y}_{1:n-1}) \right] \right] + \mathcal{C}, \quad (15)$$

*where $\mathcal{C}$ is a constant with respect to $\pi_{\boldsymbol{\theta}}$.*

The proof is provided in Appendix A.3. Since the objective $\mathcal{J}_{\tilde{W}}$ of Eq. (15) can be expressed as the sum of token-wise rewards over sampled response sequences, the entropic Wasserstein-regularized RLHF problem, Eq. (10), can be optimized using standard RL methods such as PPO (Schulman et al., 2017). The full RLHF training algorithm is provided in Algorithm 2 of Appendix B.

## 4.2 COMPUTATION OF WASSERSTEIN PENALTY

As shown in the objective of Eq. (15), computing the Wasserstein penalty requires obtaining the optimal dual solution $\phi$ of the entropic optimal transport problem. To this end, we need to compute the vectors $\mathbf{u}$ and $\mathbf{v}$ introduced in Proposition 1. These can be efficiently obtained by applying the Sinkhorn-Knopp algorithm (Sinkhorn & Knopp, 1967) for the matrix scaling problem, as described in Algorithm 1 (Cuturi, 2013; Cuturi & Doucet, 2014).

Specifically, as shown in Proposition 1, the optimal primal solution, $\boldsymbol{P}^{(n)^*}$, can be expressed as the product of the positive matrix, $\exp(-\lambda \boldsymbol{C})$, and two

---

**Algorithm 1** Computation of Wasserstein Penalty via Sinkhorn-Knopp Algorithm

---

**Input:** Learned policy $\pi_{\boldsymbol{\theta}}(\cdot|\mathbf{x}, \mathbf{y}_{1:n-1})$, Reference policy $\pi_{\text{ref}}(\cdot|\mathbf{x}, \mathbf{y}_{1:n-1})$, Cost $\boldsymbol{C}$
1: $\mathbf{u} \leftarrow \mathbf{1}_d, \mathbf{v} \leftarrow \mathbf{1}_d, \boldsymbol{K} \leftarrow \exp(-\lambda \boldsymbol{C})$
2: **while** converged **do**
3:     $\text{diag}(\mathbf{u}) \leftarrow \pi_{\boldsymbol{\theta}} ./ \boldsymbol{K}(\text{diag}(\mathbf{v}))$
4:     $\text{diag}(\mathbf{v}) \leftarrow \pi_{\text{ref}} ./ \boldsymbol{K}^\top(\text{diag}(\mathbf{u}))$
5: **end while**
6: $\phi \leftarrow -\frac{1}{\lambda} \log(\mathbf{u})$
**Output:** Dual variable $\phi$

---

diagonal matrices, $\text{diag}(\mathbf{u})$ and $\text{diag}(\mathbf{v})$, with positive entries. Since $\boldsymbol{P}^{(n)^*}$ is a transportation map, it must be doubly stochastic. Consequently, solving for $\mathbf{u}$ and $\mathbf{v}$ in Proposition 1 reduces to a matrix

scaling problem, which can be solved using the Sinkhorn-Knopp algorithm. This algorithm iteratively rescales the rows and columns of $\boldsymbol{P}^{(n)*}$ to match the target marginals $\pi_{\boldsymbol{\theta}}$ and $\pi_{\text{ref}}$, respectively (lines 3-4 in Algorithm 1, where ./ denotes element-wise division).

**Practical Consideration**   In practice, the Sinkhorn-Knopp algorithm can be directly applied, but it requires iterative matrix multiplications with the exponential of the cost matrix $\boldsymbol{K} := \exp(-\lambda \boldsymbol{C}) \in \mathbb{R}_+^{d \times d}$. This incurs $\mathcal{O}(d^2)$ computational complexity with respect to the dictionary size $d$, leading to increased time and memory consumption. To mitigate this, we employ two forms of truncation.

First, during pre-computation of the cost matrix, we apply a *nearest-$k_1$ truncation*. For each token, distances are computed only to its $k_1$ nearest neighbors. Entries outside this neighborhood are set to zero in $\boldsymbol{K}$, which is equivalent to assigning infinite distance. This yields a sparse $\boldsymbol{K}$, enabling efficient sparse matrix multiplications that reduce both storage and computation. Second, during the Sinkhorn-Knopp algorithm, we apply a *top-$k_2$ truncation*. The distributions $\pi_{\boldsymbol{\theta}}$ and $\pi_{\text{ref}}$ are truncated to their top-$k_2$ indices together with the actually sampled index, while the remaining probability mass is aggregated into a dummy index. This reduces the effective support size from $d$ to at most $2k_2 + 2$, lowering the complexity from $\mathcal{O}(d^2)$ to $\mathcal{O}(k_2^2)$. See Appendix C.3 for details of both truncations. Together, these truncations substantially reduce the computational cost of the entropic Wasserstein distance, with training time per step increasing by only 2.5% compared to standard KL regularization.

## 5   EXPERIMENTS

### 5.1   EXPERIMENTAL SETTINGS

**Tasks and Training Details**   To evaluate our Wasserstein policy regularization, we conduct open-ended text generation experiments on two datasets: the TL;DR dataset (Völske et al., 2017) for text summarization and the Anthropic Helpful and Harmless (HH-RLHF) dataset (Bai et al., 2022) for dialogue generation. We follow the experimental setup of Chai et al. (2025)[2], which provides open-source implementations for RLHF research. Our base model is the pre-trained Gemma-2B (Team et al., 2024b), and we use identical training configurations across all baselines and our method, varying only the regularization hyperparameters. For each method, the policy regularization hyperparameter $\beta$ is selected via grid search to identify the value at which training remained stable, and the best-performing model is reported. For Wasserstein policy regularization, we define the cost function as the Euclidean distance in the fixed token embedding space from the reference policy, set $\lambda = 100$, and apply truncation hyperparameters $k_1 = 512$ and $k_2 = 128$. Further experimental details are provided in Appendix C.

**Baselines**   We compare regularization based on various divergences with the proposed entropic Wasserstein-based regularization. Specifically, in addition to our approach using the entropic Wasserstein distance in Eq. (10), we evaluate reverse KL (RKL) divergence in Eq. (8), as well as token-level divergence in Eq. (7) instantiated with alternative $f$-divergences, including forward KL (FKL), JS, $\alpha$-divergence with $\alpha = 0.5$, total variation (TV), and $\chi^2$ divergence. Each $f$-divergence can be expressed in the form of a penalty on the reward through its defining function $f$, and the corresponding functions for each divergence are summarized in Table 11 of Appendix C.2.

$$\max_{\pi_{\boldsymbol{\theta}}} \mathcal{J}_f(\pi_{\boldsymbol{\theta}}; \pi_{\text{ref}}) \tag{16}$$

$$= \mathbb{E}_{\mathbf{x} \sim \mathcal{D}} \left[ \sum_{n=1}^{N} \mathbb{E}_{\mathbf{y}_n \sim \pi_{\boldsymbol{\theta}}(\mathbf{y}_n | \mathbf{x}, \mathbf{y}_{1:n-1})} \left[ R(\mathbf{x}, \mathbf{y}_{1:n}) - \beta \frac{\pi_{\text{ref}}(y_n | \mathbf{x}, \mathbf{y}_{1:n-1})}{\pi_{\boldsymbol{\theta}}(y_n | \mathbf{x}, \mathbf{y}_{1:n-1})} f\left( \frac{\pi_{\boldsymbol{\theta}}(y_n | \mathbf{x}, \mathbf{y}_{1:n-1})}{\pi_{\text{ref}}(y_n | \mathbf{x}, \mathbf{y}_{1:n-1})} \right) \right] \right].$$

**Evaluation**   We adopt GPT-4 win rate, a widely used evaluation metric in recent LLM studies (Zheng et al., 2023; Chai et al., 2025), as our primary metric. For evaluation, we randomly sample 50 validation instances and generate model responses, repeating this procedure five times. Then, GPT-4 is asked to perform pairwise comparisons between model outputs and report a win rate. We use the GPT-4 evaluation prompts from Chai et al. (2025), with the full prompt included in Appendix C.4. For TL;DR, we assess relevance, coherence, consistency, and fluency; while for HH-RLHF we focus on helpfulness. To reduce evaluation bias, we randomize the order of the responses.

---

[2] https://github.com/ernie-research/MA-RLHF

Table 1: Comparison of win rates for policy regularization with various divergences, compared to SFT and RKL-regularized PPO on the TL;DR and the HH-RLHF datasets with the Gemma-2B model.

| Divergence | TL;DR | | HH-RLHF | |
|---|---|---|---|---|
| | vs. SFT | vs. RKL | vs. SFT | vs. RKL |
| RKL | $0.848_{\pm 0.021}$ | - | $0.828_{\pm 0.010}$ | - |
| FKL | $0.316_{\pm 0.026}$ | $0.040_{\pm 0.011}$ | $0.808_{\pm 0.048}$ | $0.564_{\pm 0.019}$ |
| JS | $0.540_{\pm 0.024}$ | $0.204_{\pm 0.029}$ | $0.744_{\pm 0.031}$ | $0.424_{\pm 0.024}$ |
| $\alpha\ (\alpha = 0.5)$ | $0.724_{\pm 0.031}$ | $0.304_{\pm 0.016}$ | $0.792_{\pm 0.047}$ | $0.524_{\pm 0.046}$ |
| TV | $0.364_{\pm 0.039}$ | $0.052_{\pm 0.021}$ | $0.748_{\pm 0.038}$ | $0.376_{\pm 0.010}$ |
| $\chi^2$ | $0.904_{\pm 0.026}$ | $0.540_{\pm 0.030}$ | $0.796_{\pm 0.039}$ | $0.344_{\pm 0.017}$ |
| **Wasserstein (ours)** | $\mathbf{0.924}_{\pm 0.019}$ | $\mathbf{0.608}_{\pm 0.029}$ | $\mathbf{0.852}_{\pm 0.027}$ | $\mathbf{0.616}_{\pm 0.039}$ |

Table 2: Win rates on TL;DR with Gemma-7B. '-2B' compares to the 2B models in Table 1, and '-7B' to the 7B baselines.

| | vs. SFT-2B | vs. RKL-2B | vs. SFT-7B | vs. RKL-7B |
|---|---|---|---|---|
| RKL | **0.948** | 0.668 | 0.912 | - |
| **Wasserstein** | **0.948** | **0.712** | **0.924** | **0.532** |

Table 3: Win rates on HH-RLHF with Qwen1.5-1.8B-Chat.

| | vs. SFT | vs. RKL |
|---|---|---|
| RKL | 0.716 | - |
| **Wasserstein** | **0.752** | **0.560** |

## 5.2 QUANTITATIVE RESULTS

**Comparison to Other Divergences**    Table 1 reports the performance of summarization and dialogue generation tasks under different policy regularization methods. Additionally, Figure 2 presents win-rate comparisons against RKL across varying sampling temperatures on the HH-RLHF dataset. As the results demonstrate, our proposed Wasserstein policy regularization method achieves the best results on both datasets. In contrast, $f$-divergence-based methods rely on probability ratios between policies, which can produce exploding values and unstable training. This issue is evident in the particularly poor TL;DR results of FKL and TV. By comparison, our method remains well-defined even under support mismatch, enabling stable training and delivering superior performance consistently.

**MT-Bench Results**    To further empirically validate our approach, we evaluate model using MT-Bench, a GPT-4-based benchmark shown to strongly correlate with human preference judgments (Zheng et al., 2023). We follow the official implementation[3] and apply single-answer grading to models fine-tuned on HH-RLHF. As shown in Table 4, our method achieves the highest performance among all baselines, indicating that semantic-aware regularization improves broader conversational and instruction following abilities.

Table 4: MT-Bench score comparison on Gemma-2B trained on HH-RLHF, evaluated with GPT-4 single-answer grading.

| Divergence | Score |
|---|---|
| RKL | 4.000 |
| FKL | 4.247 |
| JS | 4.197 |
| $\alpha\ (\alpha = 0.5)$ | 4.256 |
| TV | 4.072 |
| $\chi^2$ | 4.144 |
| **Wasserstein** | **4.272** |

**Other LLM backbones**    To assess the scalability and generalization of WPR, we evaluate the method on larger and architecturally distinct LLM backbones. Using Gemma-7B on the TL;DR summarization task, Table 2 shows that WPR continues to outperform the RKL-regularized baseline. We further demonstrate generalization by training Qwen-1.5-1.8B-Chat (Bai et al., 2023) on HH-RLHF. As shown in Table 3, WPR improves performance over the RKL-regularized model. These results indicate that WPR remains consistently effective across different architectures and model scales.

**Code Generation**    We also examine WPR in a different application domain, code generation. Following the experimental settings used in Chai et al. (2025), we assess CodeGemma-7B (Team et al., 2024a) on the APPS dataset (Hendrycks et al., 2021). We compute a reward using the compiler-execution signal employed in prior work (Chai et al., 2025), and we report both the reward and the pass@1 metric over the full 5k test set. As shown in Table 5, WPR achieves consistent improvements across the Introductory, Interview, and Competition levels, as well as in the overall performance.

---

[3] https://github.com/lm-sys/FastChat/tree/main/fastchat/llm_judge

Table 5: Performance comparison on APPS with the CodeGemma-7B model.

| | Introductory | | Interview | | Competition | | All | |
|---|---|---|---|---|---|---|---|---|
| | Reward | pass@1 | Reward | pass@1 | Reward | pass@1 | Reward | pass@1 |
| SFT | 0.1024 | 23.12 | -0.1720 | 4.86 | -0.3239 | 1.48 | -0.1475 | 7.84 |
| RKL | 0.1387 | 24.00 | -0.1316 | 5.28 | -0.2910 | 1.76 | -0.1093 | 8.32 |
| **Wasserstein** | **0.1606** | **24.78** | **-0.1062** | **5.75** | **-0.2638** | **1.92** | **-0.0843** | **8.79** |

Table 6: Ablation study of WPR on TL;DR.

| Method | | Win rate | |
|---|---|---|---|
| | | vs. SFT | vs. RKL |
| Our default settings | | $0.924_{\pm 0.019}$ | $0.608_{\pm 0.029}$ |
| Cost change | (L2 → cosine) | $0.932_{\pm 0.014}$ | $0.644_{\pm 0.047}$ |
| Decreased $k_1$ | (512 → 256) | $0.920_{\pm 0.006}$ | $0.572_{\pm 0.025}$ |
| Decreased $k_2$ | (128 → 64) | $0.864_{\pm 0.015}$ | $0.528_{\pm 0.032}$ |
| Decreased $\lambda$ | (100 → 10) | $0.868_{\pm 0.024}$ | $0.552_{\pm 0.010}$ |
| Decreased Sinkhorn iterations | (10 → 5) | $0.708_{\pm 0.027}$ | $0.328_{\pm 0.026}$ |
| Increased Sinkhorn iterations | (10 → 30) | $0.880_{\pm 0.021}$ | $0.536_{\pm 0.029}$ |

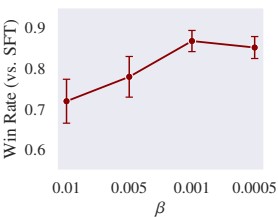

Figure 4: Sensitivity analysis of the policy regularization hyperparameter $\beta$ on HH-RLHF.

## 5.3 ANALYSIS OF WASSERSTEIN POLICY REGULARIZATION

**Ablation Study**   We conduct an ablation study to better understand the effect of components in our Wasserstein policy regularization framework, with results summarized in Table 6. Changing the cost function from the Euclidean to cosine distance yields slightly improved results, suggesting that the framework is robust to the choice of token-level cost metric. Decreasing the truncation parameters $k_1$ and $k_2$ or the entropy regularization strength $\lambda$ leads to a slight drop in performance, though our method still consistently outperforms RKL. Smaller $k_1$ and $k_2$ introduce approximation errors in the distance computation, and a smaller $\lambda$ produces overly sharp couplings that reduce stability. In practice, we use the default settings, which provide consistently robust performance across datasets and configurations. The number of Sinkhorn iterations also affects the distance computation. Reducing iterations from 10 to 5 leads to a sharp drop in performance due to insufficient convergence, while increasing iterations to 30 provides no additional benefit over the default setting. These results suggest that a moderate number of iterations is sufficient for achieving a balance between accuracy and computational efficiency.

Additionally, we analyze the sensitivity analysis of the policy regularization coefficient $\beta$ in Figure 4. Our Wasserstein-regularized approach achieves stable performance across a broad range of $\beta$ values, consistently outperforming the SFT baseline. In contrast, we observe that the $f$-divergence regularized RLHF yields stable training only within narrow ranges of $\beta$, as also reported in the previous work (Wang et al., 2024). While our method demonstrates robustness over a wider range of $\beta$, it still requires selecting an appropriate $\beta$, highlighting a fundamental limitation of RLHF. Developing approaches that reduce or remove this dependence is an important direction for future work.

**Wasserstein Penalty**   Figure 5 compares the KL and Wasserstein penalties computed during training on TL;DR, where both are scaled by the optimal regularization coefficient $\beta$ and jointly normalized to the range $[0, 1]$ using a shared min-max range. Note that larger penalties correspond to greater deviation from the reference policy. As shown in the figure, the two penalties exhibit a strong positive correlation, with a Pearson correlation coefficient of 0.917. This result demonstrates that our Wasserstein penalty, similar to the KL penalty, increases as the learned policy differs from the reference policy. Moreover, the fitted trend line has a slope of 0.579, which is less than 1, with a substantial fraction of points lying below the line, indicating that the Wasserstein penalty tends to be more lenient than KL.

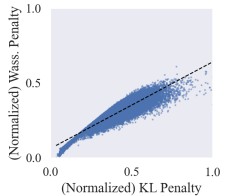

Figure 5: Normalized KL vs. Wasserstein penalty.

Table 7: Pearson correlation between each negative penalty and BERTScore (Zhang et al., 2020).

| | TL;DR | HH-RLHF |
|---|---|---|
| KL penalty | 0.1734 | 0.0172 |
| **Wasserstein penalty** | **0.2160** | **0.1749** |

Table 8: Semantic coherence of top-10 token candidates on each dataset.

| | TL;DR | HH-RLHF |
|---|---|---|
| RKL | $3.781 \pm 0.005$ | $3.690 \pm 0.004$ |
| **Wasserstein** | $\mathbf{3.593} \pm 0.003$ | $\mathbf{3.584} \pm 0.004$ |

(a) Normalized token-wise KL vs. Wasserstein penalties

(b) 2nd token: `fairs`

(c) 11th token: `Fair`

Figure 6: Case study of penalties on Gemma-2B. The prompt is "*What fair is the largest fair in Massachusetts?*", and the generated response is "*The largest fairs in Massachusetts include: 1. Boston Fair: ...*". (a) Normalized penalties for each generated token. (b-c) Next-token distribution from each policy, along with the relevant cost matrix entries, for the 2nd and 11th tokens highlighted in (a).

To understand how the Wasserstein penalty captures semantic relationships and influences model behavior, we conduct the following analyses. We first evaluate whether the penalty aligns with semantic similarity using BERTScore (Zhang et al., 2020). For responses generated by the reference and learned policies, we compute BERTScore with averaged KL and Wasserstein penalties. Because higher BERTScore indicates greater semantic similarity, we correlate it with the negative value of each penalty. As shown in Table 7, the Wasserstein penalty shows a stronger positive correlation with BERTScore, providing quantitative evidence that WPR better reflects semantic similarity.

We further analyze how the penalty behaves in actual LLM distributions. For the example in Figure 6a, KL fluctuates widely, whereas WPR often assigns minimal penalty. To further investigate, we directly compare the next-token distributions at specific tokens. For semantically similar substitutions in Figure 6b, KL assigns a large penalty due to an exact index mismatch, while WPR gives a small penalty by recognizing semantic proximity. Conversely, when probability mass shifts toward unrelated tokens in Figure 6c, WPR assigns a large penalty, correctly signaling semantic drift.

We also measure semantic coherence of the learned LLMs. For every generated token, we extract the top-10 next-token candidates and compute their mean pairwise embedding distance; smaller distances indicate greater semantic coherence. As shown in Table 8, WPR consistently produces more semantically coherent candidate sets than KL, with statistically significant margins.

Together, these results show that WPR penalizes semantic drift and promotes coherent semantic structure. We conjecture that this semantic awareness contributes to the improved alignment performance.

## 6 CONCLUSION

In this work, we propose a semantic-aware policy regularization framework for RLHF based on the entropy-regularized Wasserstein distance, which captures semantic similarity between tokens beyond the limits of KL and other $f$-divergences. By formulating the regularization in the dual space, our method yields tractable penalties compatible with standard RL algorithms, while remaining computationally efficient via the Sinkhorn-Knopp algorithm. Experiments on summarization and dialogue generation tasks demonstrate consistent improvements over KL- and $f$-divergence-based baselines, with higher win rates and MT-Bench score. These results highlight the effectiveness of semantic-aware policy distances for stable and robust alignment of large language models.

ACKNOWLEDGMENTS

This work was supported by the InnoCORE program of the Ministry of Science and ICT (N10260008) (50%). This work was supported by the IITP (Institute of Information & Communications Technology Planning & Evaluation)-ITRC (Information Technology Research Center) grant funded by the Korea government (Ministry of Science and ICT) (IITP-2026-RS-2024-00437268) (50%).

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

# A PROOFS AND DERIVATIONS

## A.1 DERIVATION OF EQ. (13)

We derive the dual problem in Eq. (13) from the Lagrangian $\mathcal{L}$ in Eq. (12) constructed for the entropic primal transportation problem in Eq. (11). First, we rewrite the Lagrangian as follows:

$$\mathcal{L}(\boldsymbol{P}^{(n)}, \boldsymbol{\phi}, \boldsymbol{\psi}) \tag{17}$$

$$:= \sum_{i=1}^{d} \sum_{j=1}^{d} \left( P_{ij}^{(n)} C_{ij} + \frac{1}{\lambda} P_{ij}^{(n)} (\log P_{ij}^{(n)} - 1) \right) \tag{18}$$

$$+ \sum_{i=1}^{d} \phi_i([\pi_{\boldsymbol{\theta}}]_i - \sum_{k=1}^{d} P_{ik}^{(n)}) + \sum_{j=1}^{d} \psi_j([\pi_{\text{ref}}]_j - \sum_{k=1}^{d} P_{kj}^{(n)})$$

$$= \sum_{i=1}^{d} \sum_{j=1}^{d} P_{ij}^{(n)} \left( C_{ij} - \phi_i - \psi_j + \frac{1}{\lambda}(\log P_{ij}^{(n)} - 1) \right) + \sum_{i=1}^{d} \phi_i([\pi_{\boldsymbol{\theta}}]_i) + \sum_{j=1}^{d} \psi_j([\pi_{\text{ref}}]_j), \tag{19}$$

where $\{\phi_i\}_{i=1}^{d}$ and $\{\psi_j\}_{j=1}^{d}$ are the Lagrange multipliers. Based on this Lagrangian, the primal and dual problem can be written as follows:

$$(\text{Primal}) \quad \min_{\boldsymbol{P}^{(n)}} \max_{\boldsymbol{\phi}, \boldsymbol{\psi}} \mathcal{L}(\boldsymbol{P}^{(n)}, \boldsymbol{\phi}, \boldsymbol{\psi}), \tag{20}$$

$$(\text{Dual}) \quad \max_{\boldsymbol{\phi}, \boldsymbol{\psi}} \min_{\boldsymbol{P}^{(n)}} \mathcal{L}(\boldsymbol{P}^{(n)}, \boldsymbol{\phi}, \boldsymbol{\psi}). \tag{21}$$

By differentiating the Lagrangian with respect to $P_{ij}^{(n)}$, we derive the condition that the optimal $P_{ij}^{(n)*}$ satisfy as follows:

$$C_{ij} - \phi_i - \psi_j + \frac{1}{\lambda} \log P_{ij}^{(n)*} = 0 \quad \Leftrightarrow \quad P_{ij}^{(n)*} = \exp\left(\lambda(\phi_i + \psi_j - C_{ij})\right). \tag{22}$$

Therefore, by substituting the optimal $P_{ij}^{(n)*}$ in Eq. (22) into Eq. (19), we can express the dual problem of Eq. (21) as

$$\max_{\phi, \psi} \sum_{i=1}^{d} \phi_i[\pi_{\boldsymbol{\theta}}]_i + \sum_{j=1}^{d} \psi_j[\pi_{\text{ref}}]_j - \sum_{i=1}^{d} \sum_{j=1}^{d} \frac{1}{\lambda} \exp(\lambda(\phi_i + \psi_j - C_{ij})). \tag{23}$$

## A.2 PROOF OF PROPOSITION 1

**Proposition 1.** (*Cuturi & Doucet, 2014*) *There exists a pair of vectors* $(\mathbf{u}, \mathbf{v}) \in \mathbb{R}_+^d \times \mathbb{R}_+^d$ *such that the optimal solutions of* $\boldsymbol{P}^{(n)}$, $\boldsymbol{\phi}$, *and* $\boldsymbol{\psi}$ *are respectively given by*

$$\boldsymbol{P}^{(n)*} = \text{diag}(\mathbf{u}) \exp(-\lambda \boldsymbol{C}) \text{diag}(\mathbf{v}), \quad \boldsymbol{\phi}^* = -\frac{1}{\lambda} \log(\mathbf{u}), \quad \boldsymbol{\psi}^* = -\frac{1}{\lambda} \log(\mathbf{v}). \tag{14}$$

*Proof.* From Eq. (22) in the derivation of Appendix A.1, the optimal $P_{ij}^{(n)*}$ can be written as

$$P_{ij}^{(n)*} = \exp\left(\lambda(\phi_i + \psi_j - C_{ij})\right) = \exp(\lambda\phi_i) \exp(-\lambda C_{ij}) \exp(\lambda\psi_j). \tag{24}$$

Defining the positive kernel $\boldsymbol{K} := \exp(-\lambda \boldsymbol{C})$, where the exponential is applied element-wise, and the scaling vectors $\mathbf{u} := \exp(\lambda\boldsymbol{\phi})$ and $\mathbf{v} := \exp(\lambda\boldsymbol{\psi})$, the optimal coupling admits the compact representation

$$\boldsymbol{P}^{(n)} = \text{diag}(\mathbf{u}) \, \boldsymbol{K} \, \text{diag}(\mathbf{v}). \tag{25}$$

Since $\boldsymbol{K}$ is strictly positive, the Sinkhorn-Knopp theorem (Sinkhorn & Knopp, 1967) guarantees the existence and uniqueness (up to an additive constant) of strictly positive scaling vectors $\mathbf{u}, \mathbf{v} \in \mathbb{R}_+^d$ such that $\boldsymbol{P}^{(n)} \in U_n(\pi_{\boldsymbol{\theta}}, \pi_{\text{ref}})$. Hence, the primal optimum $\boldsymbol{P}^{(n)}$ is unique and necessarily of the form $\text{diag}(\mathbf{u}) \, \boldsymbol{K} \, \text{diag}(\mathbf{v})$.

Finally, by the definitions of $\mathbf{u}$ and $\mathbf{v}$, the corresponding optimal dual variables are given by

$$\boldsymbol{\phi}^* = -\frac{1}{\lambda} \log \mathbf{u}, \qquad \boldsymbol{\psi}^* = -\frac{1}{\lambda} \log \mathbf{v}, \tag{26}$$

which yields the stated representation

$$\boldsymbol{P}^{(n)^*} = \mathrm{diag}(\mathbf{u}) \, \exp(-\lambda \boldsymbol{C}) \, \mathrm{diag}(\mathbf{v}), \qquad \boldsymbol{\phi}^* = -\frac{1}{\lambda} \log \mathbf{u}, \qquad \boldsymbol{\psi}^* = -\frac{1}{\lambda} \log \mathbf{v}. \tag{27}$$

$\square$

## A.3 Proof of Theorem 2

**Theorem 2.** *Let $\boldsymbol{\phi}^*(\mathbf{x}, \mathbf{y}_{1:n-1})$ denote the optimal dual variables of the entropic optimal transport problem in Eq. (13). Then, the entropic Wasserstein-regularized RLHF in Eq. (10) can be equivalently written as a reward maximization problem with an additional penalty, induced by $\boldsymbol{\phi}^*$, i.e.,*

$$\mathcal{J}_{\tilde{W}}(\pi_{\boldsymbol{\theta}}; \pi_{ref}) = \mathbb{E}_{\mathbf{x} \sim \mathcal{D}} \left[ \sum_{n=1}^{N} \mathbb{E}_{y_n \sim \pi_{\boldsymbol{\theta}}(y_n|\mathbf{x}, \mathbf{y}_{1:n-1})} \left[ R(\mathbf{x}, \mathbf{y}_{1:n}) - \beta \phi_{y_n}^*(\mathbf{x}, \mathbf{y}_{1:n-1}) \right] \right] + \mathcal{C}, \tag{15}$$

*where $\mathcal{C}$ is a constant with respect to $\pi_{\boldsymbol{\theta}}$.*

*Proof.* First, the objective of Wasserstein-regularized RLHF can be written as

$$\mathcal{J}_{\tilde{W}}(\pi_{\boldsymbol{\theta}}; \pi_{\mathrm{ref}}) := \tag{28}$$

$$\mathbb{E}_{\mathbf{x} \sim \mathcal{D}} \left[ \sum_{n=1}^{N} \mathbb{E}_{y_n \sim \pi_{\boldsymbol{\theta}}(y_n|\mathbf{x}, \mathbf{y}_{1:n-1})} \left[ R(\mathbf{x}, \mathbf{y}_{1:n}) \right] - \beta \sum_{n=1}^{N} D_{\tilde{W}}^{\lambda} \left( \pi_{\boldsymbol{\theta}}(y_n|\mathbf{x}, \mathbf{y}_{1:n-1}) || \pi_{\mathrm{ref}}(y_n|\mathbf{x}, \mathbf{y}_{1:n-1}) \right) \right].$$

By strong duality, the entropic Wasserstein distance $D_{\tilde{W}}^{\lambda}$ is equal to the optimal objective value of the dual problem in Eq. (13). Substituting the optimal solutions from Proposition 1, we obtain

$$D_{\tilde{W}}^{\lambda} \left( \pi_{\boldsymbol{\theta}}(y_n|\mathbf{x}, \mathbf{y}_{1:n-1}) || \pi_{\mathrm{ref}}(y_n|\mathbf{x}, \mathbf{y}_{1:n-1}) \right) \tag{29}$$

$$= \sum_{i=1}^{d} \phi_i^*(\mathbf{x}, \mathbf{y}_{1:n-1}) \pi_{\boldsymbol{\theta}}(y_n = i|\mathbf{x}, \mathbf{y}_{1:n-1}) + \sum_{j=1}^{d} \psi_j^*(\mathbf{x}, \mathbf{y}_{1:n-1}) \pi_{\mathrm{ref}}(y_n = j|\mathbf{x}, \mathbf{y}_{1:n-1})$$

$$- \sum_{i=1}^{d} \sum_{j=1}^{d} \frac{1}{\lambda} \exp(\lambda(\phi_i^*(\mathbf{x}, \mathbf{y}_{1:n-1}) + \psi_j^*(\mathbf{x}, \mathbf{y}_{1:n-1}) - C_{ij})) \tag{30}$$

$$= \sum_{i=1}^{d} \phi_i^*(\mathbf{x}, \mathbf{y}_{1:n-1}) \pi_{\boldsymbol{\theta}}(y_n = i|\mathbf{x}, \mathbf{y}_{1:n-1}) + \mathcal{C} \tag{31}$$

$$= \mathbb{E}_{y_n \sim \pi_{\boldsymbol{\theta}}(y_n|\mathbf{x}, \mathbf{y}_{1:n-1})} \left[ \phi_{y_n}^*(\mathbf{x}, \mathbf{y}_{1:n-1}) \right] + \mathcal{C}, \tag{32}$$

where $\mathcal{C}$ denotes a constant with respect to $\pi_{\boldsymbol{\theta}}$.

Substituting Eq. (32) into Eq. (28), the objective reduces to a reward maximization problem with an additional penalty induced by the dual variables $\boldsymbol{\phi}^*$:

$$\mathcal{J}_{\tilde{W}}(\pi_{\boldsymbol{\theta}}; \pi_{\mathrm{ref}})$$

$$= \mathbb{E}_{\mathbf{x} \sim \mathcal{D}} \left[ \sum_{n=1}^{N} \mathbb{E}_{y_n \sim \pi_{\boldsymbol{\theta}}(y_n|\mathbf{x}, \mathbf{y}_{1:n-1})} \left[ R(\mathbf{x}, \mathbf{y}_{1:n}) \right] - \beta \sum_{n=1}^{N} \mathbb{E}_{y_n \sim \pi_{\boldsymbol{\theta}}(y_n|\mathbf{x}, \mathbf{y}_{1:n-1})} \left[ \phi_{y_n}^*(\mathbf{x}, \mathbf{y}_{1:n-1}) \right] + \mathcal{C} \right] \tag{33}$$

$$= \mathbb{E}_{\mathbf{x} \sim \mathcal{D}} \left[ \sum_{n=1}^{N} \mathbb{E}_{y_n \sim \pi_{\boldsymbol{\theta}}(y_n|\mathbf{x}, \mathbf{y}_{1:n-1})} \left[ R(\mathbf{x}, \mathbf{y}_{1:n}) - \beta \phi_{y_n}^*(\mathbf{x}, \mathbf{y}_{1:n-1}) \right] \right] + \mathcal{C} \tag{34}$$

$\square$

# B  TRAINING ALGORITHM OF RLHF WITH WASSERSTEIN POLICY REGULARIZATION

---

**Algorithm 2** RLHF with Wasserstein Policy Regularization

---

**Input:** Current policy $\pi_{\boldsymbol{\theta}}$, Old policy $\pi_{\boldsymbol{\theta}^-}$, Reference policy $\pi_{\mathrm{ref}}$, Reward model $r(\mathbf{x}, \mathbf{y})$, Cost matrix $\boldsymbol{C}$, Dataset $\mathcal{D}$
1: **for** $t_{\mathrm{train}} = 1$ **to** $T_{\mathrm{train}}$ **do**
2:     Sample $\mathbf{y}_{1:d} \sim \pi_{\boldsymbol{\theta}}(\cdot|\mathbf{x})$ for $\mathbf{x} \sim \mathcal{D}$ (Computation with Batch Samples)
3:     Get $R(\mathbf{x}, \mathbf{y}_{1:n})$ with reward model $r$ for $n = \{1, 2, \cdots, d\}$
4:     Compute $\phi_{y_n}^*(\mathbf{x}, \mathbf{y}_{1:n-1})$ via Algorithm 1 using $\boldsymbol{C}$ for $n = \{1, 2, \cdots, d\}$
5:     Obtain $\hat{A}_n$ for $n = \{1, 2, \cdots, d\}$ via Eq. (35)
6:     Compute $\nabla_{\boldsymbol{\theta}} \mathcal{J}_{\tilde{\mathrm{W}}}(\boldsymbol{\theta})$ via Eq. (40)
7:     Compute $\nabla_{\boldsymbol{\psi}} \mathcal{L}_V(\boldsymbol{\psi})$ via Eq. (38)
8:     Update $\boldsymbol{\theta} \leftarrow \boldsymbol{\theta} + \eta_\pi \nabla_{\boldsymbol{\theta}} \mathcal{J}_{\tilde{\mathrm{W}}}(\boldsymbol{\theta})$, $\boldsymbol{\psi} \leftarrow \boldsymbol{\psi} - \eta_V \nabla_{\boldsymbol{\psi}} \mathcal{L}_V(\boldsymbol{\psi})$ and $\boldsymbol{\theta}^- \leftarrow \boldsymbol{\theta}$
9: **end for**
**Output:** Learned policy $\pi_{\boldsymbol{\theta}}$

---

In this section, we present the detailed training algorithm for RLHF with Wasserstein Policy Regularization (WPR). As in conventional RLHF (Ouyang et al., 2022), we iteratively sample response as $y_n \sim \pi_{\boldsymbol{\theta}^-}(\mathbf{x}, \mathbf{y}_{1:n-1})$ to get $\mathbf{y}_{1:d}$. Here, $\pi_{\boldsymbol{\theta}^-}$ is old policy whose parameters $\boldsymbol{\theta}^-$ are periodically updated by that of the current policy, $\pi_{\boldsymbol{\theta}}$. At each token generation step $n$, we adopt Generalized Advantage Estimation (GAE) (Schulman et al., 2016) for penalized reward $R'(\mathbf{x}, \mathbf{y}_{1:n}) = R(\mathbf{x}, \mathbf{y}_{1:n}) - \beta \phi_{y_n}^*(\mathbf{x}, \mathbf{y}_{1:n-1})$ in Eq. (15).

Then, the advantage with GAE denoted as $\hat{A}_n$ at each step $n$ can be expressed as

$$\hat{A}_n = \sum_{l \geq 0} (\gamma\lambda)^l \, \delta_{n+l}, \tag{35}$$

where

$$\delta_n = R'(\mathbf{x}, \mathbf{y}_{1:n}) + \gamma V_{\boldsymbol{\psi}}(\mathbf{x}, \mathbf{y}_{1:n}) - V_{\boldsymbol{\psi}}(\mathbf{x}, \mathbf{y}_{1:n-1}). \tag{36}$$

Here, $\gamma$ is a discount factor; $\lambda$ is a hyperparameter for GAE; and $V_{\boldsymbol{\psi}}$ is a value network, which estimates the discounted cumulative reward or return of given state $(\mathbf{x}, \mathbf{y}_{1:n})$, denoted as $\hat{G}_n$. Thus, the learning loss for $V_{\boldsymbol{\psi}}$ is defined as

$$\mathcal{L}_V(\boldsymbol{\psi}) = \mathbb{E}_{\mathbf{x} \sim \mathcal{D}} \left[ \sum_{n=1}^{N} \mathbb{E}_{y_n \sim \pi_{\boldsymbol{\theta}^-}(y_n|\mathbf{x}, \mathbf{y}_{1:n-1})} \left[ (V_{\boldsymbol{\psi}}(\mathbf{x}, \mathbf{y}_{1:n}) - \hat{G}_n)^2 \right] \right]. \tag{37}$$

Then, its gradient is expressed as follows.

$$\nabla_{\boldsymbol{\psi}} \mathcal{L}_V(\boldsymbol{\psi}) = \mathbb{E}_{\mathbf{x} \sim \mathcal{D}} \left[ \sum_{n=1}^{N} \mathbb{E}_{y_n \sim \pi_{\boldsymbol{\theta}^-}(y_n|\mathbf{x}, \mathbf{y}_{1:n-1})} \left[ 2(V_{\boldsymbol{\psi}}(\mathbf{x}, \mathbf{y}_{1:n}) - \hat{G}_n) \, \nabla_{\boldsymbol{\psi}} V_{\boldsymbol{\psi}}(\mathbf{x}, \mathbf{y}_{1:n}) \right] \right] \tag{38}$$

In RLHF, $V_{\boldsymbol{\psi}}$ and $\pi_{\boldsymbol{\theta}}$ are updated together. By substituting a step-wise penalized reward $R'(\mathbf{x}, \mathbf{y}_{1:n})$ with $\hat{A}_n$, Eq. (15) is expressed as

$$\mathcal{J}_{\tilde{\mathrm{W}}}(\boldsymbol{\theta}) = \mathbb{E}_{\mathbf{x} \sim \mathcal{D}} \left[ \sum_{n=1}^{N} \mathbb{E}_{y_n \sim \pi_{\boldsymbol{\theta}^-}(y_n|\mathbf{x}, \mathbf{y}_{1:n-1})} \left[ \frac{\pi_{\boldsymbol{\theta}}}{\pi_{\boldsymbol{\theta}^-}} \hat{A}_n \right] \right] + \mathcal{C}. \tag{39}$$

Here, we denote $\mathcal{J}_{\tilde{\mathrm{W}}}(\pi_{\boldsymbol{\theta}}; \pi_{\boldsymbol{\theta}^-}, \pi_{\mathrm{ref}})$ as $\mathcal{J}_{\tilde{\mathrm{W}}}(\boldsymbol{\theta})$ for simplicity and $\frac{\pi_{\boldsymbol{\theta}}}{\pi_{\boldsymbol{\theta}^-}}$ is an importance weight. Then, the gradient of $\mathcal{J}_{\tilde{\mathrm{W}}}$ is computed as

$$\nabla_{\boldsymbol{\theta}} \mathcal{J}_{\tilde{\mathrm{W}}}(\boldsymbol{\theta}) = \mathbb{E}_{\mathbf{x} \sim \mathcal{D}} \left[ \sum_{n=1}^{N} \mathbb{E}_{y_n \sim \pi_{\boldsymbol{\theta}^-}(y_n|\mathbf{x}, \mathbf{y}_{1:n-1})} \left[ (\frac{\pi_{\boldsymbol{\theta}}}{\pi_{\boldsymbol{\theta}^-}} \hat{A}_n) \nabla_{\boldsymbol{\theta}} \log \pi_{\boldsymbol{\theta}} \right] \right]. \tag{40}$$

In practice, clipping mechanism for advantage computation (Schulman et al., 2017) is adopted for Eq. (39) and $V_{\psi}(\mathbf{x}, \mathbf{y}_{1:n-1})$ in Eq. (38) is also clipped for a stable training. Algorithm 2 presents the overall training framework for RLHF with Wasserstein Policy Regularization. In Algorithm 2, $t_{\text{train}}$ and $T_{\text{train}}$ are the training step and the maximum training step, respectively. At $t_{\text{train}} = 1$, current policy $\pi_{\boldsymbol{\theta}}$, old policy $\pi_{\boldsymbol{\theta}-}$, and reference policy $\pi_{\text{ref}}$ are all initialized with SFT model. In Line # 8 in Algorithm 2, we update $\boldsymbol{\theta}^{-}$ with updated $\boldsymbol{\theta}$ at every training step, and $\eta_{\pi}$ and $\eta_{V}$ are learning rate for $\pi_{\boldsymbol{\theta}}$ and $V_{\boldsymbol{\psi}}$, respectively.

## C ADDITIONAL EXPERIMENTAL SETTINGS

### C.1 DATASETS

**TL;DR** For the summarization task, the policy is trained to generate concise summaries of Reddit posts. The dataset includes 93K preference pairs for training and 86K pairs for validation. Training data is derived from the Reddit TL;DR corpus (Völske et al., 2017). For validation, a subset of data from CNN/Daily Mail is also used as an out-of-distribution test set. The dataset is downloaded from Hugging Face.[4]

**HH-RLHF** For dialogue generation, we use the Anthropic HH-RLHF dataset (Bai et al., 2022), where the policy is trained to produce responses that are both helpful and harmless in single-turn and multi-turn dialogue settings. It comprises 112K preference-labeled instances for training and an additional 12.5K instances for validation. The dataset can be downloaded from Hugging Face.[5]

**APPS** For code generation, we use the APPS dataset (Hendrycks et al., 2021), which provides a diverse set of programming problems requiring executable Python solutions. The dataset consists of 5K training instances and 5K validation instances, each containing a natural language problem dedscription paired with unit tests for automated evaluation. The dataset can be downloaded from Hugging Face.[6]

### C.2 MODEL TRAINING DETAILS

We follow the experimental setup of Chai et al. (2025), which provides open-source implementations for RLHF research.[7] This implementation is based on the Deepspeed-Chat package (Yao et al., 2023), and we adopt its configuration as the default setting. Our base model is the pre-trained Gemma-2B (Team et al., 2024b), and we use identical training configurations across all baselines and our method, varying only the regularization hyperparameters. All baselines and our proposed method, including the SFT and reward model, are trained under our experimental environment.

**Supervised Fine-Tuning (SFT)** We split each dataset into three subsets and allocate 20% for supervised fine-tuning. Prompts are paired with their preferred responses to construct instruction data. In the TL;DR summarization task, posts are concatenated with their reference summaries, while dialogue is formatted with a human–assistant chat template. For this stage, we employ the Gemma-2B model as the backbone. The training configuration specifies a batch size of 512, a learning rate of $5 \times 10^{-5}$, a cosine learning rate scheduler with a warmup ratio of 0.1, and a total of 3 epochs.

**Reward Model Training** InstructGPT (Ouyang et al., 2022) mitigates distributional mismatch by fine-tuning the reward model on the same dataset used for SFT. Following this approach, we also train our reward model on the identical dataset. In this stage, 40% of the data is used for reward model training. Preference annotations are processed in the same way as in SFT. The reward model is initialized from the SFT checkpoint. The training configuration specifies a batch size of 64, a learning rate of $1 \times 10^{-5}$, a cosine learning rate scheduler with a warmup ratio of 0.1, and a single epoch.

For code generation, we follow prior work (Chai et al., 2025) and use a reward function derived directly from the compiler execution singal, without training an additional reward model. We adopt

---

[4] https://huggingface.co/datasets/openai/summarize_from_feedback
[5] https://huggingface.co/datasets/Dahoas/full-hh-rlhf
[6] https://huggingface.co/datasets/codeparrot/apps
[7] https://github.com/ernie-research/MA-RLHF

Table 9: Hyperparameters for PPO training.

| Hyperparameter | Value |
|---|---|
| PPO epochs | 1 |
| Rollout | 1 |
| Clip ratio | 0.2 |
| $\lambda$ in GAE | 0.95 |
| $\gamma$ in GAE | 1 |
| Max prompt length | 512 |
| Max response length | 256 |
| Warmup steps | 200 |
| Temperature | 0.8 |
| Top-p | 1.0 |
| Top-k | 50 |

Table 10: Policy regularization hyperparameter $\beta$ for each method.

| Divergence | TL;DR | HH-RLHF |
|---|---|---|
| RKL | 0.005 | 0.001 |
| FKL | 0.05 | 0.0001 |
| JS | 0.05 | 0.01 |
| $\alpha$ ($\alpha = 0.5$) | 0.01 | 0.05 |
| TV | 0.01 | 0.01 |
| $\chi^2$ | 0.001 | 0.001 |
| **Wasserstein** | 0.05 | 0.0005 |

Table 11: Corresponding functions for each $f$-divergences.

| Divergence | $f(u)$ |
|---|---|
| RKL | $u \log u$ |
| FKL | $-\log u$ |
| JS | $u \log u - (u+1) \log(\frac{u+1}{2})$ |
| $\alpha$ | $\frac{1}{\alpha(\alpha-1)}(u^{1-\alpha} - (1-\alpha)u - \alpha)$ |
| TV | $\frac{1}{2}|u-1|$ |
| $\chi^2$ | $(u-1)^2$ |

the adaptive compiler-based reward used in previous studies (Chai et al., 2025; Shojaee et al., 2023; Liu et al., 2023). For a generated solution $\mathbf{y}$ to a problem $\mathbf{x}$, the reward is defined as:

$$r(\mathbf{x}, \mathbf{y}) = \begin{cases} -0.3 + 1.3 \cdot \dfrac{N_{\text{pass}}}{N_{\text{pass}} + N_{\text{fail}}}, & \text{if } y \text{ compiles successfully,} \\ -0.6, & \text{if } y \text{ raises a runtime error,} \\ -1.0, & \text{if } y \text{ fails to compile.} \end{cases}$$

Here, $N_{\text{pass}}$ and $N_{\text{fail}}$ denote the number of unit tests passed and failed, respectively.

**Policy Optimization with PPO** The remaining 40% of the dataset is used for PPO training. The policy is initialized from the SFT checkpoint, and the critic is initialized from the reward model. We use the same SFT and reward model checkpoints for all baselines. The training configuration uses a batch size of 256, learning rates of $1.5 \times 10^{-5}$ for both the policy and the critic, and runs for one epoch. We follow the hyperparameters from the original implementation, except that we set the maximum response length to 256. The hyperparameters are summarized in Table 9.

**Policy Regularization** For each method, the policy regularization hyperparameter $\beta$ is selected via grid search to identify the value at which training remained stable, and we report the best-performing model. Specifically, we perform a grid search over $\{0.5, 0.1, 0.05, 0.01, 0.005, 0.001, 0.0005, 0.0001\}$. The resulting $\beta$ values used for each baseline are summarized in Table 10. Each $f$-divergence can be expressed in the form of a penalty on the reward through its defining function $f$, and the corresponding functions for each divergence are summarized in Table 11.

$$\max_{\pi_{\boldsymbol{\theta}}} \mathcal{J}_f(\pi_{\boldsymbol{\theta}}; \pi_{\text{ref}}) \tag{41}$$

$$= \mathbb{E}_{\mathbf{x} \sim \mathcal{D}} \left[ \sum_{n=1}^{N} \mathbb{E}_{\mathbf{y}_n \sim \pi_{\boldsymbol{\theta}}(\mathbf{y}_n | \mathbf{x}, \mathbf{y}_{1:n-1})} \left[ R(\mathbf{x}, \mathbf{y}_{1:n}) - \beta \frac{\pi_{\text{ref}}(y_n | \mathbf{x}, \mathbf{y}_{1:n-1})}{\pi_{\boldsymbol{\theta}}(y_n | \mathbf{x}, \mathbf{y}_{1:n-1})} f\left( \frac{\pi_{\boldsymbol{\theta}}(y_n | \mathbf{x}, \mathbf{y}_{1:n-1})}{\pi_{\text{ref}}(y_n | \mathbf{x}, \mathbf{y}_{1:n-1})} \right) \right] \right].$$

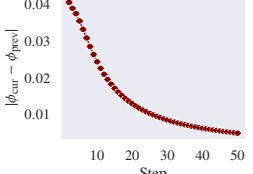

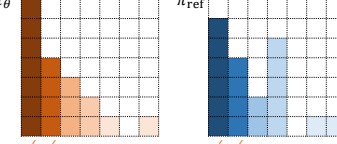

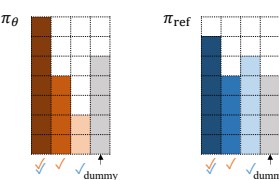

Figure 7: Convergence of the Sinkhorn-Knopp algorithm.

Figure 8: Example of top-$k_2$ truncation where $k_2 = 2$.

For WPR, we define the cost function as the Euclidean distance in the fixed token embedding space from the SFT model, set $\lambda = 100$, and apply truncation hyperparameters $k_1 = 512$ and $k_2 = 128$. The number of Sinkhorn iterations is set to 10 for TL;DR and 50 for HH-RLHF.

### C.3 SHINKORN ALGORITHM DETAILS

**Stopping Criterion** The Sinkhorn-Knopp iterations involve alternating updates of the scaling vectors $\mathbf{u}$ and $\mathbf{v}$. Since the dual variable $\phi$ is ultimately used as the regularization penalty, we monitor convergence based on the change in $\phi = -\frac{1}{\lambda} \log \mathbf{u}$. Iterations are terminated when the change in $\phi$ falls below a pre-defined tolerance, which we use $10^{-4}$. For practicality, we also impose a maximum number of Sinkhorn iterations, as specified in Appendix C.2.

**Numerical Stability** The Sinkhorn updates involve repeated rescaling operations and log computations when recovering $\phi$, which can lead to numerical instabilities. To mitigate this, we add small constants to denominators and log arguments. Figure 7 plots the evolution of the convergence metric across iterations on Gemma-2B experiments with HH-RLHF, demonstrating that the truncated Sinkhorn procedure converges stably in practice.

**Nearest-$k_1$ Truncation for Cost Matrix** Since the cost matrix $C$ is fixed throughout training, we pre-compute $K = \exp(-\lambda C)$. However, storing the full $K$ is infeasible for vocabularies of extremely large tokens. To address this, for each token we retain only its $k_1$ nearest neighbors and set all other entries to zero, which is equivalent to assigning infinite cost to distant tokens. We additionally enforce symmetry by mirroring retained entries so that the sparse kernel remains consistent. This sparsification enables efficient sparse-matrix multiplications during Sinkhorn iterations.

**Top-$k_2$ Truncation for Token Distributions** For computational efficiency, we also truncate the token distributions of both the target policy and the reference policy. Specifically, we retain the top-$k_2$ probability indices along with the index of the sampled token. The remaining probability mass is aggregated into a dummy index. Because the sampled token is always included, the required dual variable $\pi^*_{y_n}$ can be recovered. A conceptual illustration is provided in Figure 8.

### C.4 EVALUATION DETAILS

We adopt GPT-4 win rate, a widely used evaluation metric in recent LLM studies (Zheng et al., 2023; Chai et al., 2025), as our main evaluation measure. For each comparison, we randomly sample 50 validation instances and generate model responses, repeating this procedure five times. Unless otherwise noted, the sampling temperature is fixed at 0.5. Then, GPT-4 is asked to perform pairwise comparisons between model outputs and compute the win rate. We use the `gpt-4o-2024-05-13` model for all evaluations. We follow the GPT-4 evaluation prompts provided by Chai et al. (2025), and for completeness, we include the full prompt below. For TL;DR, we assess relevance, coherence, consistency, and fluency; while for HH-RLHF we focus on helpfulness. To reduce evaluation bias, we randomize the order of the responses.

---

**GPT-4 Evaluation Prompt for TL;DR**

You will be given two summaries written for an article. Your task is to pick the better one between them, based on the four criteria. Please make sure you read and understand these instructions very carefully.

Relevance - selection of important content from the source. The summary should include only important information from the source document. Annotators were instructed to penalize summaries that contained redundancies and excess information.

Coherence - the collective quality of all sentences. We align this dimension with the DUC quality question of structure and coherence whereby "the summary should be well-structured and well-organized. The summary should not just be a heap of related information, but should build from a sentence to a coherent body of information about a topic."

Consistency - the factual alignment between the summary and the summarized source. A factually consistent summary contains only statements that are entailed by the source document. Annotators were also asked to penalize summaries that contained hallucinated facts.

Fluency - the quality of the summary in terms of grammar, spelling, punctuation, word choice, and sentence structure.

You should output a single character to indicate which summary you think is better. 'A' stands for Summary A and 'B' stands for Summary B. If you think both summaries are equally good, output 'E'.

Article:
{article}

Summary A:
{summary_a}

Summary B:
{summary_b}

Your Choice (only a single character, you are allowed to think both summaries are equal and output 'E'):

---

**GPT-4 Evaluation Prompt for HH-RLHF**

For the following query to a chatbot assistant, which response is more helpful?

First provide a one-sentence comparison of the two responses and explain which you feel is more helpful. Second, on a new line, state only 'A' or 'B' to indicate which response is more helpful. If they are equally good or bad, state 'E'. Your response should use the json format, with "comparison" and "choice" as keys.

Query: dialogue

Response A: {resp_a}

Response B: {resp_b}

Your Judgment:

---

Table 12: Win rates on TL;DR using the Gemma-based models, varying the policy backbone and the embedding spaces used to form the cost matrix $C$.

| Method | Backbone | Embedding | Win rate (vs. SFT-2B) | Win rate (vs. RKL-2B) |
|---|---|---|---|---|
| RKL-regularized PPO | Gemma-2B | - | 0.848 | - |
| | Gemma-7B | - | **0.948** | 0.668 |
| **Wasserstein-regularized PPO** | Gemma-2B | Gemma-2B | 0.924 | 0.608 |
| | | Gemma-7B | 0.908 | 0.556 |
| | Gemma-7B | Gemma-2B | 0.944 | 0.684 |
| | | Gemma-7B | **0.948** | **0.712** |

Table 13: Sensitivity analysis of the truncation hyperparameter $k_2$ on TL;DR with Gemma-2B. *Time* is the wall-clock time for the penalty computation.

| $k_2$ | Time (hours/1k steps) | Win rate (vs. SFT) | Win rate (vs. RKL) |
|---|---|---|---|
| 64 | 0.08 | 0.864 | 0.528 |
| 128 | 0.12 | 0.924 | 0.608 |
| 256 | 0.19 | 0.916 | 0.584 |

## D    ADDITIONAL EXPERIMENTAL RESULTS

### D.1    ANALYSIS OF EMBEDDING SPACE

The semantic cost matrix $C$ is constructed from token embeddings and plays a central role in WPR. Since the cost must be computed over the full vocabulary of the policy model, the embedding space must be aligned with its tokenizer. Therefore, only models sharing the same tokenizer can be used directly. Using embeddings from a model with a different tokenizer would require building a cross-token alignment, a promising but nontrivial direction for future work.

To study the effect of embedding quality, we conduct an experiment using Gemma-2B and Gemma-7B, which share the same tokenizer. For each model, we extract the frozen token embeddings after SFT and used them to construct the cost matrix $C$. We then independently varied: (1) the policy backbone (Gemma-2B or Gemma-7B), and the embedding source used to form $C$ (Gemma-2B SFT or Gemma-7B SFT).

As shown in Table 12, across all configurations, WPR consistently outperforms RKL-regularized PPO when using the same policy backbone, indicating that WPR provides benefits regardless of the specific embedding model used. As expected, the policy backbone size has the largest effect on performance, with Gemma-7B outperforming Gemma-2B.

Interestingly, constructing $C$ using Gemma-7B embeddings for a Gemma-2B backbone does not yield performance improvements over using the 2B embeddings. We conjecture that this is because each policy is naturally grounded in the token geometry encoded by its own SFT embedding space. Thus, the embedding space of the same model backbone is most compatible with the policy's internal representation.

### D.2    ADDITIONAL SENSITIVITY ANALYSIS

**Truncation hyperparameter** $k_2$    To evaluate how the computational overhead and performance scale with the truncation hyperparameter $k_2$, we vary $k_2 \in \{64, 128, 256\}$ and measure both the penalty computation time and the resulting win rates. The results are summarized in Table 13. Increasing $k_2$ from 64 to 128 improves performance, with a moderate increase in penalty computation time, especially small compared to the overall training time of approximately 4.5 hours per 1,000 steps. Increasing $k_2$ further from 128 to 256 yeilds minimal performance gains, and in fact slightly decreases performance. This suggests that $k_2 = 128$ already captures most of the probability mass of the token distribution and provides an accurate approximation. Accordingly, we adopt $k_2 = 128$ for all experiments in the paper.

Table 14: Sensitivity analysis of the entropy regularization parameter $\lambda$ on TL;DR with Gemma-2B.

| $\lambda$ | Win rate (vs. SFT) | Win rate (vs. RKL) |
|---|---|---|
| 50 | 0.900 | 0.564 |
| 100 | 0.924 | 0.608 |
| 200 | 0.916 | 0.612 |

Table 15: Detailed breakdown of the wall-clock time per 1,000 training steps. The time required to compute the regularization penalty differs across methods, whereas the generation and training steps are independent of the regularization method and therefore reported using unified timings.

| | Time (hours/1k steps) | |
|---|---|---|
| | RKL | WPR |
| Generation | 0.769 | |
| Penalty computation | 0.005 | 0.117 |
| Backpropagation | 3.707 | |
| Total | 4.481 | 4.593 |

**Entropy regularization hyperparameter $\lambda$**   The hyperparameter $\lambda$ controls the level of entropic smoothing in the Sinkhorn distance, determining the balance between the semantic fidelity and the smoothness of the transport plan. Smaller values of $\lambda$ place greater weight on entropy, producing overly soft transport plans and diminishing the influence of semantic structure. Conversely, excessively large values of $\lambda$ cause the kernel $\boldsymbol{K} = \exp(-\lambda \boldsymbol{C})$ to collapse toward zero, creating numerical oscillations during the Sinkhorn rescaling steps.

To the best of our knowledge, this work is the first to apply an entropy-regularized Wasserstein penalty in the token space of LLMs during RL fine-tuning. Therefore, we initially selected $\lambda$ empirically and found that $\lambda = 100$ provided the stable and consistent performance across tasks. We use this value in all experiments reported in the paper. We additionally provide a sensitivity analysis with $\lambda \in \{50, 100, 200\}$. As shown in Table 14, WPR consistently outperforms RKL-based regularization for all tested values (with win rate is greater than 0.5). As expected, smaller values reduce the influence of semantic structure and lead to a performance drop, consistent with the interpretation above.

### D.3   COMPUTATIONAL RESOURCES

**Wall-clock Time**   Table 15 reports a detailed breakdown of the wall-clock time per 1,000 PPO training steps for RKL and WPR. The measurement is decomposed into (1) generation, (2) penalty computation, and (3) backpropagation. For this report, we use 4 A100 GPUs using the Gemma-2B policy model with the TL;DR dataset, employing 8 batches per GPU and 8 gradient accumulation steps.

WPR requires additional computation during the penalty step because it incorporates semantic structure across tokens, whereas the KL penalty incurs nearly zero overhead. However, the added cost is minor relative to the forward and backward passes of a billion-parameter LLM. Note that the generation and backpropagation stages remain identical across regularization methods, and their runtimes are influenced far more by the generated response length, and therefore we report unified timings for these stages.

**GPU Memory Usage**   The primary memory overhead of WPR arises from the cost matrix $\boldsymbol{C}$. Since $\boldsymbol{C}$ is fixed across all training steps, it is computed once before training and reused throughout PPO optimization. Its memory footprint is independent of the model size.

As described in Section 4.2, we apply truncation methods that allow $\boldsymbol{C}$ to be stored as a sparse matrix, requiring approximately 12.8GB of GPU memory in our configuration. This overhead is reflected in the peak GPU memory measurement in Table 16. During training, additional memory

Table 16: Peak GPU memory usage (GB) for RKL and Wasserstein regularization, measured on a single A100 GPU with a batch size of 8.

|  | GPU usage (GB) |
| --- | --- |
| RKL | 64.05 |
| **Wasserstein** | 78.98 |

usage depends primarily on the batch size. When needed, memory can be reduced by increasing the gradient accumulation steps, at the cost of longer runtime.

## E    GENERATED EXAMPLES

We present qualitative comparisons of response generation outputs. Table 17 shows an example from the TL;DR dataset, where we compare responses generated with reverse KL regularization and with our proposed Wasserstein Policy Regularization. Similarly, Table 18 provides an example from the HH-RLHF dataset under the same comparison. These examples illustrate that WPR produces responses that are better aligned with the underlying prompt compared to RKL.

## F    LLM USAGE

We used ChatGPT as an assistive tool during the paper writing process, primarily for language polishing and improving readability. Additionally, we employed GPT-4o API in the experimental evaluation as an automatic judge to compare model responses and compute win rates, following established evaluation protocols.

Table 17: Comparison of response generation outputs from the prompt on the TL;DR dataset with RKL and WPR.

| Prompt | POST
Subreddit: r/pettyrevenge
I was driving home from work in San Antonio when I noticed this jerk in a sports car tailgating, honking, flashing lights, and in general being a major pest. We pulled up to a red light and he gunned it right through the intersection, somehow managing to not cause a major accident. I fumed for a bit but figured that was the end of it.

Ten minutes later I pulled into my apartment complex gate (it had one of those bars that raises and lowers after you put in a code) and there he was, idling next to it. I don't know if he was just visiting so didn't have the right entry numbers or what, but he was just waiting for someone to come in so he could slip by on their pass.

I pulled up to the gate, put my code in, and saw him inches from my bumper waiting to race in behind me. The gate raised up, I went forward, he followed intently, and I stopped with my rear bumper just barely in front of where the gate would fall.

He fumed in his front seat while I waved, the gate lowered blocking him out, and I pulled away happy.
TL;DR: |
|---|---|
| **Responses** | |
| **RKL** (A) | 26 year old male tailgating and harassing other motorists in San Antonio, TX causing minor fender bender when attempting to sneak past gate with illegal entry pass. Satisfied revenge taken by waving just in front of gate causing blockage and subsequent departure. |
| **WPR (ours)** (B) | jerk in sports car tailgating me, gunned it through intersection somehow avoiding major accident. Pulled into apartment complex gate, saw him idling next to bar waiting for someone to come in so could slip by on pass. Gate raised up blocking him out and I pulled away happy. |
| **GPT-4 Judgment** | B |

Table 18: Comparison of response generation outputs from the prompt on the HH-RLHF dataset with RKL and WPR.

| Prompt | Human: I need some assistance writing. See I was adopted years ago, and I am trying to meet my paternal mother. Can you write a letter asking if it is possible for us to meet?

Assistant: |
| --- | --- |
| **Responses** | |
| **RKL** (A) | Yes, absolutely! Writing a letter to your biological mother asking for the opportunity to meet is a great way to initiate the process of connecting with your birth family. Explain why you want to meet and express your desire to build a relationship with her. Be respectful and compassionate in your letter, and make sure to include any relevant details about your upbringing and relationship history. Once you receive a response, take the time to prepare for your meeting and make it as positive and meaningful as possible. |
| **WPR (ours)** (B) | Here is an example letter for requesting a meeting with your biological mother:

Dear Mother,

I would like to request a meeting to reconnect with you after many years of separation. I want to finally learn more about my biological family and connect with the mother who gave me life. Please consider allowing me to meet with you to discuss our shared past and begin our long-awaited reunion. Thank you. |
| **GPT-4 Judgment** | Response B is more helpful because it provides a concrete example of a letter, which directly addresses the user's request. |

