# OpenReview forum: "Semantic-aware Wasserstein Policy Regularization for Large Language Model Alignment"
_ICLR.cc/2026/Conference — ICLR 2026 Poster_

### Official Review · Reviewer_AZq3 · 2025-10-24

**Soundness:** 2
**Presentation:** 2
**Contribution:** 2
**Rating:** 4
**Confidence:** 3

**Summary:**

This paper proposes Wasserstein Policy Regularization (WPR), a novel semantic-aware regularization framework for RLHF. Instead of using classical KL or f-divergence, the authors propose using an entropy-regularized Wasserstein distance that captures token-level semantic similarity in the embedding space. Theoretical derivations (dual formulation and tractable penalty) are rigorous, and experiments on Gemma-2B across two tasks (TL;DR summarization and HH-RLHF dialogue) show consistent improvements over KL- and f-divergence-based baselines.

**Strengths:**

1. The paper is theoretically rigorous and clearly presents the primal–dual equivalence and the functional role of the dual potential in policy regularization.
2. The experimental design is clear and sound, with proper variable control and consistent setups across baselines. WPR demonstrates robust and superior performance across multiple tasks and hyperparameter settings.
3. The idea is conceptually novel and insightful, providing a solution to alleviate the limitation of the traditional KL divergence in measuring semantically similar but support-disjoint samples.

**Weaknesses:**

Major:
1. The paper’s core claim is that the Wasserstein distance captures semantic relationships between tokens, but this claim is not empirically validated beyond a toy example. It would substantially strengthen the work to include quantitative metrics.
2. The experiments are mainly conducted on a single base model (Gemma-2B) and two tasks. To support claims of generality and robustness, the study should include results from larger or alternative model architectures.
3. While the framework is conceptually novel, the current experiments mainly demonstrate performance gains, not semantic effects.
Quantitative analysis showing **how semantic structures influence model behavior** would significantly raise the paper’s impact.


Minor:
1. The implementation of the Sinkhorn algorithm should be described in greater detail, for example
- Stopping criterion and convergence conditions
- Truncation scheme (top-*k₁*, *k₂*) and numerical stability.
2. The evaluation heavily depends on GPT-4 win rates, a widely used but subjective metric that offers limited robustness. From my own experience, the outcome of such evaluations can vary depending on how the GPT judging prompt is designed.
If feasible, incorporating **small-scale double-blind human evaluations** could further enhance credibility. (This is only a suggestion.)

**Questions:**

Overall, the paper is well-written. However, the central concept of “semantic awareness”—which distinguishes this work from prior RLHF regularization—remains vague and unclear to me. Specifically,

1. Could you clarify the mechanism by which semantic awareness translates into improved alignment outcomes?  In tasks where the target distribution may diverge semantically from the base model.(e.g., filtering harmful or biased responses), does Wasserstein regularization still provide benefits, or could it hinder optimization?

2. Could you provide quantitative measures to illustrate the model’s semantic sensitivity?  (e.g. Is "semantically close" means high embedding correlation or something else?)

---

> ### Author Response · Authors · 2025-11-21
> **Official Comment by Authors for Reviewer AZq3 (1/4)**
>
> We appreciate the reviewer’s helpful comments and suggestions. Point-by-point responses are provided below. We have updated the manuscript to reflect the reviewer’s feedback and that of the other reviewers, and all changes in the revised version are highlighted in blue.
>
> ***
> > **Q1. [Quantitative measure for semantic relationship]** (Weaknesses Major 1) The paper’s core claim is that the Wasserstein distance captures semantic relationships between tokens, but this claim is not empirically validated beyond a toy example. It would substantially strengthen the work to include quantitative metrics.
>
> > (Questions 2) Could you provide quantitative measures to illustrate the model’s semantic sensitivity? (e.g. Is "semantically close" means high embedding correlation or something else?)
>
> Thank you for the insightful suggestion. Our detailed response is provided below, and we have incorporated the corresponding analysis into Section 5.3 (page 10) of the revised manuscript.
>
> In our work, semantically close refers to responses that carry similar meaning, rather than exact token-level agreement. For example, in the prompt “What is in this image?” for a small cat image, “cat” and “kitten” are semantically similar responses, whereas “cat” and “table” are not, as illustrated by the motivating example in Figure 1. To validate this notion beyond the toy example, we conduct an additional analysis to examine whether the proposed Wasserstein penalty exhibits this behavior in practice.
>
> As a quantitative measure of semantic similarity, we use BERTScore (Zhang et al., ICLR 2020), which, consistent with our motivation, evaluates the semantic equivalence using cosine similarity in a pre-trained embedding space rather than relying on exact token matching.
>
> In our analysis, we generate responses from both the reference policy and the learned policy. For each response pair, we compute the corresponding BERTScore. Also, we compute the per-token KL penalty and Wasserstein penalty, and average each across tokens to obtain a response-level penalty. Because higher BERTScore indicates greater semantic similarity, while lower penalties indicate greater similarity, we compare the correlation of BERTScore with the negative value of each penalty. As shown in the below table, the negative Wasserstein penalty exhibits a stronger positive Pearson correlation with BERTScore than the KL penalty.
>
> These results provide empirical evidence that the Wasserstein penalty better reflects semantic similarity, supporting our claim that WPR acts as a semantic-aware regularizer.
>
> [**Table 7 in the revised manuscript**: Pearson correlation between each negative penalty and BERTScore.]
>
> | Penalty | TL;DR  | HH-RLHF |
> |------------------------------|--------|---------|
> | KL | 0.1734 | 0.0172  |
> | Wasserstein | **0.2160** | **0.1749** |
>
> (Zhang et al., ICLR 2020) BERTScore: Evaluating Text Generation with BERT.

---

> ### Author Response · Authors · 2025-11-21
> **Official Comment by Authors for Reviewer AZq3 (2/4)**
>
> > **Q2. [Experiments on other LLM backbones]** (Weaknesses Major 2) The experiments are mainly conducted on a single base model (Gemma-2B) and two tasks. To support claims of generality and robustness, the study should include results from larger or alternative model architectures.
>
> To assess the generalization and scalability of WPR beyond Gemma-2B, we conduct additional experiments across larger models, different LLM families, and different tasks:
> * **[Larger model]** Gemma-7B on the summarization task with TL;DR dataset
> * **[Different LLM family]** Qwen-1.5-1.8B-Chat on the dialogue generation task with HH-RLHF dataset
> * **[Different task]** CodeGemma-7B on the code generation task with APPS dataset
>
> We include this analysis and the results in “Other LLM backbones” and “Code Generation” paragraphs of Section 5.2 of the revised manuscript.
>
> As shown in the below tables (Table 2, 3, and 5 of the revised manuscript), WPR consistently outperforms the RKL-regularized baseline across larger model scales, different model families, and diverse tasks. These results demonstrate that WPR is not limited to Gemma-2B and remains effective and robust across architecture. These findings highlight the robustness and general applicability of WPR.
>
> [**Table 2 in the revised manuscript**: Win rates on TL;DR with Gemma-7B. ‘-2B’ compares to the 2B models in Table 1 in the main manuscript, and ‘-7B’ to the 7B baselines.]
> | | vs. SFT-2B | vs. RKL-2B | vs. SFT-7B | vs. RKL-7B |
> |--------------|------------|------------|------------|------------|
> | RKL          | **0.948**  | 0.668      | 0.912      | -          |
> | **Wasserstein**  | **0.948**  | **0.712**  | **0.924**      | **0.532**  |
>
> [**Table 3 in the revised manuscript**: Win rates on HH-RLHF with Qwen1.5-1.8B-Chat.]
> | | vs. SFT | vs. RKL |
> |--------------|---------|---------|
> | RKL          | 0.716   | -       |
> | **Wasserstein**  | **0.752**   | **0.560**   |
>
> [**Table 5 in the revised manuscript**: Performance comparison on APPS with the CodeGemma-7B model.]
> | | Introductory Reward/pass@1  | Interview Reward/pass@1 | Competition Reward/pass@1 | All Reward/pass@1 |
> |--------------|----------------|---------------------|---------------------|------------------------|
> | SFT          | 0.1024 / 23.12          | -0.1720 / 4.86                | -0.3239 / 1.48                   | -0.1475/ 7.84          |
> | RKL          | 0.1387 / 24.00          | -0.1316 / 5.28                | -0.2910 / 1.76                   | -0.1093/ 8.32          |
> | **Wasserstein** | **0.1606** / **24.78**      | **-0.1062** / **5.75**            | **-0.2638** / **1.92**               | **-0.0843** / **8.79**      |

---

> ### Author Response · Authors · 2025-11-21
> **Official Comment by Authors for Reviewer AZq3 (3/4)**
>
> > **Q3. [Semantic awareness and LLM alignment]** (Weaknesses Major 3) While the framework is conceptually novel, the current experiments mainly demonstrate performance gains, not semantic effects. Quantitative analysis showing how semantic structures influence model behavior would significantly raise the paper’s impact.
>
> > (Questions 1) Could you clarify the mechanism by which semantic awareness translates into improved alignment outcomes? In tasks where the target distribution may diverge semantically from the base model.(e.g., filtering harmful or biased responses), does Wasserstein regularization still provide benefits, or could it hinder optimization?
>
>
> **[Case study: token-level behavior of the penalty]**
>
> To examine how semantic structure influences model behavior under WPR, we conduct a case study analyzing how the penalty is applied to the LLM distributions. We use an example: Prompt - ”What fair is the largest fair in Massachusetts?”, Response – “The largest fairs in Massachusetts include: 1. Boston Fair: This annual fair attracts thousands …”. We visualize the token-wise KL and Wasserstein penalties across the generated response (both jointly normalized to the range [0,1] using a shared min-max range).
>
> As shown in Figure 6 in the revised manuscript, the Wasserstein penalties are often negligible cost, whereas the KL penalties fluctuate widely. To understand this difference, we compare the actual next-token distributions of the reference and learned policies at specific tokens, plotting the top-k probabilities.
>
> * **(Left: semantically related shifts)** “fair” and “fairs” have nearly identical meaning, yet KL assigns a large penalty due to the mismatch at the exact token index. In contrast, the Wasserstein penalty remains small because it accounts for the semantic proximity in the embedding space.
> * **(Right: semantically unrelated shifts)** In cases where probability mass shifts occur among semantically unrelated tokens such as “winter”, “fair”, and “common”, the Wasserstein penalty becomes large, correctly signaling semantic drift.
>
> This qualitative analysis shows that WPR tolerates meaning-preserving deviations but strongly penalizes changes that alter semantic content.
>
> **[Quantitative evaluation: semantic coherence of top-k predictions]**
>
> To test whether this semantic sensitivity influences model behavior more broadly, we measure the semantic coherence of next-token predictions across 2,000 test prompts using models trained on HH-RLHF. For every generated token, we extract the model’s top-10 next token candidates and computed the mean pairwise L2 distance among their token embeddings. Smaller distances indicate a more semantically coherent candidate set. As shown in Table 8, WPR produces top-k candidate sets that are more semantically coherent than those produced under RKL, which is statistically significant.
>
> [**Table 8 in the revised manuscript**: Semantic coherence of top-10 token candidates on each dataset.]
> | | TL;DR     | HH-RLHF      |
> |--------------|---------------------------|----------------------------|
> | RKL          | 3.781 ± 0.005             | 3.690 ± 0.004             |
> | **Wasserstein**  | **3.593** ± 0.003         | **3.584** ± 0.004         |
>
>
> Together, these results show that WPR penalizes semantic drift and encourages coherent semantic structure in the token prediction distribution. This likely helps prevent unnecessary changes when the learned policy is already semantically close to the reference policy, allowing the model to maintain meaning-preserving behavior while still permitting reward-driven shifts when required. This may contribute to the improved alignment performance observed in our experiments.
>
> **[Diverge target distribution]**
>
> Regarding tasks where the target distribution intentionally diverges semantically from the base model, our observations suggest that WPR does not meaningfully hinder optimization. WPR encourages the learned policy to stay closer to the reference policy under a distance metric that reflects semantic similarity between tokens, but it remains a regularization term. When the reward model strongly penalizes harmful responses, the reward dominates the update, and the policy can move sufficiently far from the reference. In other words, WPR shapes the geometry of the distance being minimized, but the degree of divergence is still governed by the reward signal.
>
> Empirically, the HH-RLHF dataset contains harmful/harmless distinctions, and WPR still outperforms RKL on this task. This suggests that incorporating semantic structure into the regularizer does not hinder the model’s ability to make reward-aligned semantic shifts when necessary.

---

> ### Author Response · Authors · 2025-11-21
> **Official Comment by Authors for Reviewer AZq3 (4/4)**
>
> > **Q4. [Details of Sinkhorn algorithm implementation]** (Weaknesses Minor 1) The implementation of the Sinkhorn algorithm should be described in greater detail, for example: Stopping criterion and convergence conditions, Truncation scheme (top-k₁, k₂) and numerical stability.
>
> Thank you for the suggestion. In the revised manuscript, we have added a detailed description of the Sinkhorn algorithm implementation in Appendix C.3, addressing the points raised by the reviewer. Specifically, we now describe the following:
>
> **[Stopping criterion and convergence conditions]**
>
> Because the dual variable $\phi$ is ultimately used as our penalty term, we monitor convergence based on the change in $\phi=-\frac{1}{\lambda} \log u$, where $u$ is the scaling vector in the Sinkhorn algorithm. For practicality, we also impose a maximum number of Sinkhorn iterations, which was already specified in the original manuscript.
>
> **[Numerical stability]**
>
> We provide plots in Figure 7 of the revised manuscript showing the evolution of the convergence metric across Sinkhorn iterations, demonstrating stable convergence. In practice, the Sinkhorn updates involve repeated rescaling operations and log computations when recovering $\phi$. To avoid numerical instabilities, we add small constants to denominators and log arguments.
>
> **[Truncation scheme (nearest-$k_1$)]**
>
> Since the cost matrix $C$ is fixed, we can pre-compute the matrix $K$ before training. However, storing the full $K$ is infeasible due to the large vocabulary size. To address this, for each token we retain only its $k_1$ nearest neighbors and set all other entries to zero. This is equivalent to assigning infinite cost to distant tokens, and it keeps $K$ sparse. We also enforce symmetry by ensuring that whenever a entry is retained in one direction, the corresponding entry in the opposite direction is also kept. This enables efficient sparse-matrix multiplications in the Sinkhorn updates.
>
> **[Truncation scheme (top-$k_2$)]**
>
> For both the target and reference policy, we retain only the top-$k_2$ probability indices plus the index of the sampled token. The remaining probability mass is mapped to a dummy index that accumulates all discarded mass. This truncated distribution is then used within the Sinkhorn updates. Because the sampled token index is always included, the required dual potential $\phi^*_{y_n}$ can still be obtained. A detailed illustration is included in Figure 8 of the revised manuscript.
>
> ***
> > **Q5. [Human evaluations]** (Weaknesses Minor 2) The evaluation heavily depends on GPT-4 win rates, a widely used but subjective metric that offers limited robustness. From my own experience, the outcome of such evaluations can vary depending on how the GPT judging prompt is designed. If feasible, incorporating small-scale double-blind human evaluations could further enhance credibility. (This is only a suggestion.)
>
> We appreciate the reviewer’s concern about the GPT-based evaluation. We agree that GPT-4 win rates, while widely used in recent LLM studies, inevitably rely on an automated judge and may be sensitive to prompt design. As explained in the Evaluation paragraph of Section 5.1, we use the same judging prompts as Chai et al. (2025) without any modification.
>
> We also agree that human evaluation would further strengthen the credibility of the results. However, conducting human studies requires formal IRB approval for human-subject research, which may not be completed within the rebuttal period. We plan to incorporate human evaluations into future extensions of this research.
>
> Additionally, as noted in our response to Q2, we also evaluate WPR on a code generation task, where performance is assessed using compiler-based signals rather than GPT judgements, and WPR again outperforms the RKL baseline. This provides complementary evidence that the improvements are not solely dependent on GPT-based evaluation.

---

### Official Review · Reviewer_Ndnt · 2025-10-31

**Soundness:** 3
**Presentation:** 3
**Contribution:** 3
**Rating:** 4
**Confidence:** 3

**Summary:**

This paper introduces Wasserstein Policy Regularization (WPR), a new semantic-aware regularization for RLHF framework based on the entropy-regularized Wasserstein distance. The experiments show WPR's outperformance to other baselines such as KL-divergence and f-divergence.

**Strengths:**

1. The paper is well-written and easy to read.
2. For the regularization, both theoretical formulation and practical implementation are introduced in detail and analyzed via complexity perspective.
3. The comprehensive experiments show the outperformance of new regularization and the effect of each component in the framework.

**Weaknesses:**

1. The only base model in the experiments is the pre-trained Gemma-2B. The results of models from other LLM families will further validate the effectiveness of the regularization.
2. The authors introduce the ignorance of semantic similarity in the previous regularization as the main motivation, and highlight that proposed WPR is a semantic-aware regularization. However, the analysis about semantic awareness is missed in the description of the regularization method. In the experiments, this is no corresponding results to prove that WPR captures semantic relationships (like the motivating example in Figure 1). I think the further discussion focusing on semantic awareness is needed if it is the key for outperformance.

I would like to adjust my score if these concerns are addressed.

**Questions:**

1. For the cost matrix $\mathbf{C}$, is it constant? If it is constant, could it be possible to use some matrix decomposition such as SVD to speed up the computation of $\exp(-\lambda\mathbf{C})$. If it is not, how is it initialized and updated?

---

> ### Author Response · Authors · 2025-11-21
> **Official Comment by Authors for Reviewer Ndnt (1/3)**
>
> We sincerely appreciate the reviewer’s constructive and careful comments. We address each of your comments in detail below. Revisions informed by the reviewer’s feedback, as well as comments from the other reviewers, have been incorporated into the manuscript, with all modifications shown in blue.
> ***
> > **Q1. [Experiments on other LLM backbones]** (Weaknesses 1) The only base model in the experiments is the pre-trained Gemma-2B. The results of models from other LLM families will further validate the effectiveness of the regularization.
>
> To assess the generalization and scalability of WPR beyond Gemma-2B, we conduct additional experiments across larger models, different LLM families, and different tasks:
> * **[Larger model]** Gemma-7B on the summarization task with TL;DR dataset
> * **[Different LLM family]** Qwen-1.5-1.8B-Chat on the dialogue generation task with HH-RLHF dataset
> * **[Different task]** CodeGemma-7B on the code generation task with APPS dataset
>
> We include this analysis and the results in “Other LLM backbones” and “Code Generation” paragraphs of Section 5.2 of the revised manuscript.
>
> As shown in the below tables (Table 2, 3, and 5 of the revised manuscript), WPR consistently outperforms the RKL-regularized baseline across larger model scales, different model families, and diverse tasks. These results demonstrate that WPR is not limited to Gemma-2B and remains effective and robust across architecture. These findings highlight the robustness and general applicability of WPR.
>
> [**Table 2 in the revised manuscript**: Win rates on TL;DR with Gemma-7B. ‘-2B’ compares to the 2B models in Table 1 in the main manuscript, and ‘-7B’ to the 7B baselines.]
> | | vs. SFT-2B | vs. RKL-2B | vs. SFT-7B | vs. RKL-7B |
> |--------------|------------|------------|------------|------------|
> | RKL          | **0.948**  | 0.668      | 0.912      | -          |
> | **Wasserstein**  | **0.948**  | **0.712**  | **0.924**      | **0.532**  |
>
> [**Table 3 in the revised manuscript**: Win rates on HH-RLHF with Qwen1.5-1.8B-Chat.]
> | | vs. SFT | vs. RKL |
> |--------------|---------|---------|
> | RKL          | 0.716   | -       |
> | **Wasserstein**  | **0.752**   | **0.560**   |
>
> [**Table 5 in the revised manuscript**: Performance comparison on APPS with the CodeGemma-7B model.]
> | | Introductory Reward/pass@1  | Interview Reward/pass@1 | Competition Reward/pass@1 | All Reward/pass@1 |
> |--------------|----------------|---------------------|---------------------|------------------------|
> | SFT          | 0.1024 / 23.12          | -0.1720 / 4.86                | -0.3239 / 1.48                   | -0.1475/ 7.84          |
> | RKL          | 0.1387 / 24.00          | -0.1316 / 5.28                | -0.2910 / 1.76                   | -0.1093/ 8.32          |
> | **Wasserstein** | **0.1606** / **24.78**      | **-0.1062** / **5.75**            | **-0.2638** / **1.92**               | **-0.0843** / **8.79**      |

---

> ### Author Response · Authors · 2025-11-21
> **Official Comment by Authors for Reviewer Ndnt (2/3)**
>
> > **Q2. [Analysis of semantic awareness]** (Weaknesses 2) The authors introduce the ignorance of semantic similarity in the previous regularization as the main motivation, and highlight that proposed WPR is a semantic-aware regularization. However, the analysis about semantic awareness is missed in the description of the regularization method. In the experiments, this is no corresponding results to prove that WPR captures semantic relationships (like the motivating example in Figure 1). I think the further discussion focusing on semantic awareness is needed if it is the key for outperformance.
>
> Thank you for the insightful suggestion. Our detailed response is provided below, and we have incorporated the corresponding analysis into Section 5.3 (page 10) of the revised manuscript.
>
> **[Quantitative measure for semantic relationship]**
>
> In our work, semantically close refers to responses that carry similar meaning, rather than exact token-level agreement. For example, in the prompt “What is in this image?” for a small cat image, “cat” and “kitten” are semantically similar responses, whereas “cat” and “table” are not, as illustrated by the motivating example in Figure 1. To validate this notion beyond the toy example, we conduct an additional analysis to examine whether the proposed Wasserstein penalty exhibits this behavior in practice.
>
> As a quantitative measure of semantic similarity, we use BERTScore (Zhang et al., ICLR 2020), which, consistent with our motivation, evaluates the semantic equivalence using cosine similarity in a pre-trained embedding space rather than relying on exact token matching.
>
> In our analysis, we generate responses from both the reference policy and the learned policy. For each response pair, we compute the corresponding BERTScore. Also, we compute the per-token KL penalty and Wasserstein penalty, and average each across tokens to obtain a response-level penalty. Because higher BERTScore indicates greater semantic similarity, while lower penalties indicate greater similarity, we compare the correlation of BERTScore with the negative value of each penalty. As shown in the below table, the negative Wasserstein penalty exhibits a stronger positive Pearson correlation with BERTScore than the KL penalty.
>
> These results provide empirical evidence that the Wasserstein penalty better reflects semantic similarity, supporting our claim that WPR acts as a semantic-aware regularizer.
>
> [**Table 7 in the revised manuscript**: Pearson correlation between each negative penalty and BERTScore.]
>
> | Penalty | TL;DR  | HH-RLHF |
> |------------------------------|--------|---------|
> | KL | 0.1734 | 0.0172  |
> | Wasserstein | **0.2160** | **0.1749** |
>
> (Zhang et al., ICLR 2020) BERTScore: Evaluating Text Generation with BERT.
>
> **[Semantic awareness and LLM alignment]**
>
> *[Case study: token-level behavior of the penalty]*
>
> To examine how semantic structure influences model behavior under WPR, we conduct a case study analyzing how the penalty is applied to the LLM distributions. We use an example: Prompt - ”What fair is the largest fair in Massachusetts?”, Response – “The largest fairs in Massachusetts include: 1. Boston Fair: This annual fair attracts thousands …”. We visualize the token-wise KL and Wasserstein penalties across the generated response (both jointly normalized to the range [0,1] using a shared min-max range).
>
> As shown in Figure 6 in the revised manuscript, the Wasserstein penalties are often negligible cost, whereas the KL penalties fluctuate widely. To understand this difference, we compare the actual next-token distributions of the reference and learned policies at specific tokens, plotting the top-k probabilities.
>
> * **(Figure 6b: semantically related shifts)** “fair” and “fairs” have nearly identical meaning, yet KL assigns a large penalty due to the mismatch at the exact token index. In contrast, the Wasserstein penalty remains small because it accounts for the semantic proximity in the embedding space.
> * **(Figure 6c: semantically unrelated shifts)** In cases where probability mass shifts occur among semantically unrelated tokens such as “winter”, “fair”, and “common”, the Wasserstein penalty becomes large, correctly signaling semantic drift.
>
> This qualitative analysis shows that WPR tolerates meaning-preserving deviations but strongly penalizes changes that alter semantic content.
>
> (continued in the next comment)

---

> ### Author Response · Authors · 2025-11-21
> **Official Comment by Authors for Reviewer Ndnt (3/3)**
>
> (response continued)
>
> *[Quantitative evaluation: semantic coherence of top-k predictions]*
>
> To test whether this semantic sensitivity influences model behavior more broadly, we measure the semantic coherence of next-token predictions across 2,000 test prompts using models trained on HH-RLHF. For every generated token, we extract the model’s top-10 next token candidates and computed the mean pairwise L2 distance among their token embeddings. Smaller distances indicate a more semantically coherent candidate set. As shown in Table 8, WPR produces top-k candidate sets that are more semantically coherent than those produced under RKL, which is statistically significant.
>
> [**Table 8 in the revised manuscript**: Semantic coherence of top-10 token candidates on each dataset.]
> | | TL;DR     | HH-RLHF      |
> |--------------|---------------------------|----------------------------|
> | RKL          | 3.781 ± 0.005             | 3.690 ± 0.004             |
> | **Wasserstein**  | **3.593** ± 0.003         | **3.584** ± 0.004         |
>
>
> Together, these results show that WPR penalizes semantic drift and encourages coherent semantic structure in the token prediction distribution. This likely helps prevent unnecessary changes when the learned policy is already semantically close to the reference policy, allowing the model to maintain meaning-preserving behavior while still permitting reward-driven shifts when required. This may contribute to the improved alignment performance observed in our experiments.
>
> ***
> > **Q3. [Cost matrix $C$]** (Questions 1) For the cost matrix C, is it constant? If it is constant, could it be possible to use some matrix decomposition such as SVD to speed up the computation of exp(-\lambda C). If it is not, how is it initialized and updated?
>
> **[Is $C$ constant?]**
>
> In our experiments $C$ is fixed. As described in Section 5.1, we construct $C$ using the frozen token embeddings of the SFT model. Therefore, $C$ remains fixed throughout RL fine-tuning.
>
> **[Can matrix decompositions accelerate $\exp(-\lambda C)$?]**
>
> The term $\exp (-\lambda C)$ is element-wise exponential, not a matrix exponential. Therefore, matrix decompositions may not provide computational benefit for the computation of $\exp(-\lambda C)$. We currently mention this only in Appendix A (below Eq. (24)), and we clarify this explicitly in the revised main manuscript.
>
> **[Update $C$ during training]**
>
> Updating $C$ would require recomputing all pairwise distances between tokens at every training step. This is computationally intractable, so we keep the embedding space fixed.

---

### Official Review · Reviewer_rU8j · 2025-11-02

**Soundness:** 2
**Presentation:** 3
**Contribution:** 2
**Rating:** 6
**Confidence:** 4

**Summary:**

This paper introduces Wasserstein policy regularization (WPR) for RLHF. the authors discuss that standard KL-based regularization only compares token probabilities at identical indices, ignoring semantic similarity across tokens, which may penalize reasonable semantic deviations. The authors propose using the entropy-regularized Wasserstein (Sinkhorn) distance between next-token distributions, leveraging token embedding geometry to induce semantic-aware regularization. They derive a dual-form penalty that integrates seamlessly into PPO, maintaining tractability via Sinkhorn scaling with sparsity tricks. Their experiments on Gemma-2B across TL;DR and HH-RLHF datasets show consistent improvements over KL and other f-divergences in GPT-4 win rate and MT-Bench scores.

**Strengths:**

- By replacing KL with a Wasserstein-based penalty that reflects token embedding geometry, the method tries to captures semantic similarity (e.g., “cat” vs. “kitten”), which could lead to more natural, aligned outputs rather than over-penalizing harmless semantic variations. But there are some concerns as detailed in weaknesses.

- The paper derives a dual formulation enabling efficient Sinkhorn-based computation, making the method compatible with standard RLHF pipelines (e.g., PPO) and achieving empirical gains in practice.

**Weaknesses:**

- Why does token-space geometry matter in practice? What is lost with KL, this rationale should be strengthened. The authors show win-rates but don’t link it explicitly to semantic alignment behaviour.

- “Incorporates the geometry of the token space” this is unclear without explanation which deserves clarity. They rely on embedding-space distances, not true semantic grounding. This raises questions: Which embedding space? Frozen SFT model embeddings? Reference model embeddings? Is the embedding space stable or changing during fine-tuning?

-  Is the overhead of proposed method is scalable to larger models ? What are the memory impacts during training?

- Is it possible to get a DPO version of the proposed methods?

- How hard is it to lean phi star in 15?

- In (9), the definition of divergence depends on m, which eventually results in  an optimization variable for each n, and now the value of N changes from sample to sample, then how to utilize the same P* or any thoughts about it?

- Except additional training overhead, is there any other price to pay for the proposed technique?

**Questions:**

Please refer to weaknesses,

---

> ### Author Response · Authors · 2025-11-21
> **Official Comment by Authors for Reviewer rU8j (1/4)**
>
> We are grateful for the reviewer’s insightful and thorough comments, which helped improve the clarity and rigor of the manuscript. Our responses to each point are provided below. The revised version incorporates your feedback together with the comments from the other reviewers, with all updates marked in blue.
> ***
> > **Q1. [Semantic awareness and LLM alignment]** (Weaknesses 1) Why does token-space geometry matter in practice? What is lost with KL, this rationale should be strengthened. The authors show win-rates but don’t link it explicitly to semantic alignment behaviour.
>
> We appreciate the reviewer’s question. Our detailed response is provided below, and we have incorporated the corresponding analysis into Section 5.3 (page 10) of the revised manuscript.
>
> **[Case study: token-level behavior of the penalty]**
>
> To examine how semantic structure influences model behavior under WPR, we conduct a case study analyzing how the penalty is applied to the LLM distributions. We use an example: Prompt - ”What fair is the largest fair in Massachusetts?”, Response – “The largest fairs in Massachusetts include: 1. Boston Fair: This annual fair attracts thousands …”. We visualize the token-wise KL and Wasserstein penalties across the generated response (both jointly normalized to the range [0,1] using a shared min-max range).
>
> As shown in Figure 6 in the revised manuscript, the Wasserstein penalties are often negligible cost, whereas the KL penalties fluctuate widely. To understand this difference, we compare the actual next-token distributions of the reference and learned policies at specific tokens, plotting the top-k probabilities.
>
> * **(Figure 6b: semantically related shifts)** “fair” and “fairs” have nearly identical meaning, yet KL assigns a large penalty due to the mismatch at the exact token index. In contrast, the Wasserstein penalty remains small because it accounts for the semantic proximity in the embedding space.
> * **(Figure 6c: semantically unrelated shifts)** In cases where probability mass shifts occur among semantically unrelated tokens such as “winter”, “fair”, and “common”, the Wasserstein penalty becomes large, correctly signaling semantic drift.
>
> This qualitative analysis shows that WPR tolerates meaning-preserving deviations but strongly penalizes changes that alter semantic content.
>
> **[Quantitative evaluation: semantic coherence of top-k predictions]**
>
> To test whether this semantic sensitivity influences model behavior more broadly, we measure the semantic coherence of next-token predictions across 2,000 test prompts using models trained on HH-RLHF. For every generated token, we extract the model’s top-10 next token candidates and computed the mean pairwise L2 distance among their token embeddings. Smaller distances indicate a more semantically coherent candidate set. As shown in Table 8, WPR produces top-k candidate sets that are more semantically coherent than those produced under RKL, which is statistically significant.
>
> [**Table 8 in the revised manuscript**: Semantic coherence of top-10 token candidates on each dataset.]
> | | TL;DR     | HH-RLHF      |
> |--------------|---------------------------|----------------------------|
> | RKL          | 3.781 ± 0.005             | 3.690 ± 0.004             |
> | **Wasserstein**  | **3.593** ± 0.003         | **3.584** ± 0.004         |
>
>
> Together, these results show that WPR penalizes semantic drift and encourages coherent semantic structure in the token prediction distribution. This likely helps prevent unnecessary changes when the learned policy is already semantically close to the reference policy, allowing the model to maintain meaning-preserving behavior while still permitting reward-driven shifts when required. This may contribute to the improved alignment performance observed in our experiments.

---

> ### Author Response · Authors · 2025-11-21
> **Official Comment by Authors for Reviewer rU8j (2/4)**
>
> > **Q2. [Specification of embedding space]** (Weaknesses 2) “Incorporates the geometry of the token space” this is unclear without explanation which deserves clarity. They rely on embedding-space distances, not true semantic grounding. This raises questions: Which embedding space? Frozen SFT model embeddings? Reference model embeddings? Is the embedding space stable or changing during fine-tuning?
>
> As noted, the semantic geometry used by WPR depends on the token embedding space that defines the cost matrix $C$. As stated in Section 5.1, we use the frozen token embeddings of the SFT model which is same as the reference model.
>
> Allowing the embedding space to change during RL fine-tuning would require recomputing the cost at every training step. Therefore, changing embedding space is computationally intractable, so we use the fixed embedding space.
>
> Because this cost must be computed over the vocabulary of the policy model, the embedding space could be aligned with the tokenizer of the target model; thus, only models sharing the same tokenizer can be used in a straightforward manner. For models that do not share a tokenizer (e.g., GPT-4 embeddings), obtaining token-level embedding alignment requires constructing a cross-token mapping, which is an interesting but nontrivial direction for future work.
>
>
> To examine the effect of embedding quality on WPR, we conduct an additional experiment using Gemma-2B and Gemma-7B models, which share the same tokenizer. For each model, we use the frozen token embeddings obtained after SFT to construct the cost matrix $C$, and independently varied both the policy backbone and the embedding source.
>
> [**Table 12 in the revised manuscript**: Win rates on TL;DR using the Gemma-based models, varying the policy backbone and the embedding spaces used to form the cost matrix]
> | Method                       | Backbone  | Embedding | Win rate (vs. SFT-2B) | Win rate (vs. RKL-2B) |
> |-----------------------------|-----------|-----------|------------------------|------------------------|
> | RKL-regularized PPO     | Gemma-2B  | -         | 0.848                  | -                      |
> |                             | Gemma-7B  | -         | **0.948**              | 0.668                  |
> | **Wasserstein-regularized PPO** | Gemma-2B | Gemma-2B | 0.924                  | 0.608                  |
> |                             | Gemma-2B  | Gemma-7B  | 0.908                  | 0.556                  |
> |                             | Gemma-7B  | Gemma-2B  | 0.944                  | 0.684                  |
> |                             | Gemma-7B  | Gemma-7B  | **0.948**              | **0.712**              |
>
> The results show that, across all configurations, WPR consistently outperforms RKL-regularized PPO when using the same policy backbone, indicating that WPR provides benefits regardless of the embedding model used. The size of the policy backbone has the largest effect on the performance, with Gemma-7B outperforming Gemma-2B.
>
> Interestingly, using the Gemma-7B embedding space to construct the cost matrix for the Gemma-2B policy does not surpass the performance achieved when using the original 2B embeddings. We conjecture that this occurs because the policy model is naturally grounded in the token geometry encoded by its own SFT embedding space, therefore this embedding space is naturally compatible with that model.
>
> We include this analysis and the additional results in Appendix D.1 in the revised manuscript.

---

> ### Author Response · Authors · 2025-11-21
> **Official Comment by Authors for Reviewer rU8j (3/4)**
>
> > **Q3. [Scalability]** (Weaknesses 3) Is the overhead of proposed method is scalable to larger models ? What are the memory impacts during training?
>
> The primary memory overhead introduced by WPR stems from the cost matrix $C$. Importantly, since this matrix is fixed across all training steps, it can be computed once before training and reused throughout PPO optimization. Also, its memory usage is independent of the backbone model size.
>
> As discussed in “Practical Consideration” paragraph of Section 4.2 in the original manuscript, we apply truncation methods that allow $C$ to be stored as a sparse matrix. In our current configuration, storing this sparse matrix requires approximately 12.8GB of GPU memory. This overhead is reflected in the peak per-GPU memory usage measurements, as shown in the below table. During training, the remaining memory usage is determined by the batch size. If additional memory is required, the batch size can be reduced by increasing the gradient accumulation steps, albeit at the cost of increased runtime. This analysis is presented in Appendix D.3 of the revised manuscript.
>
> [**Table 16 in the revised manuscript**: Peak GPU memory usage (GB) for RKL and Wasserstein regularization, measured on a single A100 GPU with a batch size of 8.]
>
> | | GPU usage (GB) |
> |--------------|-----------------|
> | RKL          | 64.05           |
> | **Wasserstein**  | 78.98           |
>
> To empirically validate scalability of our proposed method, we conduct an additional experiment using a Gemma-7B backbone on the TL;DR summarization task. As shown in the below table, WPR continues to outperform the RKL-regularized baseline even at the 7B scale. This demonstrates that the method remains effective and practical for larger models. The analysis and results are added in “Other LLM backbones” paragraph of Section 5.2 in the revised version.
>
> [**Table 2 in the revised manuscript**: Win rates on TL;DR with Gemma-7B. ‘-2B’ compares to the 2B models in Table 1 in the main manuscript, and ‘-7B’ to the 7B baselines.]
> | | vs. SFT-2B | vs. RKL-2B | vs. SFT-7B | vs. RKL-7B |
> |--------------|------------|------------|------------|------------|
> | RKL          | **0.948**  | 0.668      | 0.912      | -          |
> | **Wasserstein**  | **0.948**  | **0.712**  | **0.924**      | **0.532**  |
>
> ***
> > **Q4. [DPO version]** (Weaknesses 4) Is it possible to get a DPO version of the proposed methods?
>
> We agree that connecting our method to a DPO-style formulation is an interesting direction. However, deriving a DPO analogue for Wasserstein regularization is fundamentally challenging.
>
> DPO relies on a closed-form relationship between the optimal policy and the reward model under the RKL-regularized reward maximization objective. This closed-form expression enables the conversion of the RLHF objective into a pairwise supervised objective over preference data.
>
> In contrast, with Wasserstein regularization, the RL objective forms a bi-level optimization problem: the policy optimization operates in the outer loop, while the Wasserstein regularization is defined via an inner optimal transport problem. Due to this nested structure, it is hard to find tractable closed-form expression linking rewards and optimal policies, which prevents a direct reduction to a DPO-style supervised objective on paired datasets.
>
> For these reasons, obtaining a DPO version of WPR is nontrivial, and we consider this an interesting direction for future work.
>
>
> ***
> > **Q5. [Optimal $\phi^*$]** (Weaknesses 5) How hard is it to lean phi star in 15?
>
> As described in Section 4.2, $\phi$ is obtained via the Sinkhorn-Knopp algorithm, a well-established method for solving entropically regularized optimal transport problems in closed form up to iterative matrix scaling. Its convergence properties are well-studied and theoretically guaranteed in prior work (Peyré and Cuturi, 2019). It should be noted that, unlike Wasserstein GANs, obtaining optimal dual variables $\phi^{*}$ does not need to learn through gradient-based training with a parametric network.
>
> As explained in the “Practical consideration” paragraph in Section 4.2, the main computational cost arises from repeated multiplications with the cost matrix, which is $O(d^2)$ complexity. We address this by applying two truncation strategies. With these truncations, computing $\phi^*$ becomes lightweight.
>
> (Peyré and Cuturi, 2019) Computational optimal transport: With applications to data science. Foundations and Trends in Machine Learning, 11(5-6), 355-607.

---

> ### Author Response · Authors · 2025-11-21
> **Official Comment by Authors for Reviewer rU8j (4/4)**
>
> > **Q6. [Clarification of $P^{(n)}$]** (Weaknesses 6) In (9), the definition of divergence depends on m, which eventually results in an optimization variable for each n, and now the value of N changes from sample to sample, then how to utilize the same P* or any thoughts about it?
>
> We believe the reviewer’s question appears to originate from viewing the transport plan $P^{(n)}$ as a global variable shared across samples. However, this is not the case. In our formulation, the optimal transport plan $P^{(n)}$ depends explicitly on the conditioning context $(x, y_{1:n-1})$, and therefore varies across both samples and token positions.
>
> More precisely, in Eq. (9), $P^{(n)}$ is defined as the optimal coupling between the two conditional token distributions: $\pi_{\theta} (\cdot | x, y_{1:n-1})$ and  $\pi_{ref} (\cdot | x, y_{1:n-1})$. Because these distributions change with the prefix $(x,y_{1:n-1})$, the corresponding optimal transport plan must also change. Thus, $P^{(n)}$ is not a single global matrix shared across samples. Note that we explicitly state in lines 283-284 of the original manuscript: “Here, $P^{(n)}$, $\phi$, and $\psi$ are functions of $(x, y_{1:n−1})$, but we omit their input terms for brevity unless this causes ambiguity.”. To avoid any potential misunderstanding, we revise the manuscript to make this dependency when Eq. (9) is introduced.
>
> ***
>
> > **Q7. [Additional price of WPR]** (Weaknesses 7) Except additional training overhead, is there any other price to pay for the proposed technique?
>
> Besides the additional training overhead, the proposed method requires tuning new hyperparameters associated with the Sinkhorn algorithm, such as the truncation level $k$, the entropic regularization strength $\lambda$, and the number of Sinkhorn iterations. These hyperparameters control the trade-off between computational cost and the granularity of the captured semantic structure.
>
> Although these hyperparameters introduce tuning overhead, they are necessary to compute the Wasserstein-based regularization that imparts semantic awareness, an ability that KL-based approaches fundamentally lack. In practice, we found the method to be robust across a broad range of hyperparameter values, and the default settings worked consistently across all tasks.

---

> > ### Comment · Reviewer_rU8j · 2025-11-27
> >
> > I thank the authors for the rebuttal. After reviewing the other reviewer's rebuttal and response to my questions, most of the concerns have been addressed, and I would like to maintain my current scores.

---

> > > ### Author Response · Authors · 2025-11-27
> > >
> > > We sincerely appreciate the reviewer's follow-up and are glad to hear that our clarifications addressed the concerns. Thank you for taking the time to review our work. The reviewer's comments were valuable in improving the manuscript.

---

### Official Review · Reviewer_wx4k · 2025-11-02

**Soundness:** 3
**Presentation:** 3
**Contribution:** 3
**Rating:** 8
**Confidence:** 3

**Summary:**

This paper identifies a key limitation in standard RLHF: the use of Kullback-Leibler (KL) divergence and other f-divergences for policy regularization is "semantically-blind," as it only compares token probabilities at identical indices . This means a policy that assigns high probability to "kitten" is penalized just as much as one that assigns high probability to "table," when the reference token is "cat" .

To solve this, the authors propose Wasserstein Policy Regularization (WPR), a novel, semantic-aware regularizer for RLHF based on the entropy-regularized Wasserstein (Sinkhorn) distance . The core technical contribution is an elegant and tractable formulation: by leveraging the dual of the optimal transport problem, the Wasserstein regularization term becomes a simple, token-wise penalty applied to the reward. This penalty is derived from the optimal dual variables (ϕ*), which are computed efficiently using the standard Sinkhorn-Knopp algorithm.

This approach makes the Wasserstein-based penalty directly compatible with standard RL algorithms like PPO. The authors demonstrate empirically on summarization (TL;DR) and dialogue (HH-RLHF) tasks that WPR significantly outperforms a wide range of f-divergence baselines (including RKL, FKL, JS, and χ²) in terms of GPT-4 win rate and MT-Bench scores.

**Strengths:**

Clear, Compelling Motivation: The paper's greatest strength is its crystal-clear motivation. The "cat/kitten/table" example in Figure 1 immediately and intuitively communicates the flaw in existing methods and the rationale for the new one .

Elegant and Tractable Formulation: The core technical contribution is the derivation of a tractable algorithm (WPR) from a complex theoretical concept (Wasserstein distance). The insight to use the dual optimal variables (ϕ*) as a direct reward penalty (Theorem 2) is the key that makes this practical and compatible with existing PPO-based RLHF pipelines.

Strong, Consistent Empirical Results: The proposed method (WPR) convincingly outperforms a comprehensive suite of f-divergence baselines on two standard alignment tasks (summarization and dialogue). The improvements are consistent across both GPT-4 win rates (Table 1) and the MT-Bench benchmark (Table 2).

Thorough Analysis and Ablation: The authors provide a strong set of analyses. The ablation study (Table 3) confirms the importance of the method's components (e.g., number of Sinkhorn iterations, λ). The sensitivity analysis (Figure 4) shows WPR is more stable across a wider range of the β hyperparameter than other divergences. The penalty analysis (Figure 5) provides insight, showing WPR is correlated with KL but "more lenient".

**Weaknesses:**

Cost Matrix C as a "Black Box": The entire semantic-awareness of the method hinges on the cost matrix C, which is built from the token embeddings of the reference SFT model. This is a reasonable choice, but its impact is not deeply explored. The ablation study only compares L2 vs. Cosine distance, but not the source or quality of the embeddings. The paper doesn't answer: what if the SFT model's embeddings are of poor quality? Would using embeddings from a more powerful, external model improve results?

Computational Overhead: The paper claims the overhead is low ("only 2.5%" increase in training time). However, this is after applying two aggressive truncations: k1=512 for the cost matrix and k2=128 for the policy distributions. This O(k_2^2) complexity is still per-token, per-step, compared to the O(1) cost of the KL penalty. A more detailed wall-clock time comparison and an analysis of the performance/compute trade-off (e.g., how much better does it get if k2=256?) would be beneficial.

New Hyperparameter λ: The method introduces a new, important hyperparameter λ, the entropy regularization strength, which is set to 100. The ablation study shows a performance drop when decreasing it to 10. This parameter's tuning and sensitivity are critical to the method's success, and a more in-depth discussion of its impact and how to set it would improve the paper's practical utility.

**Questions:**

On the Cost Matrix C: The semantic cost matrix C is fundamental to the method's success and is derived from the SFT model's embeddings. Have you experimented with using a different source for these embeddings, such as from a larger, more capable, and static model (e.g., GPT-4 embeddings)? Is it possible that a "better" or more semantically-grounded cost matrix C would lead to even larger gains?

On Computational Overhead: Could you please provide a more detailed breakdown of the computational overhead? The "2.5%" figure is encouraging, but a wall-clock time comparison per 1000 training steps (or per-epoch) against the RKL baseline would be very informative. How much does this overhead scale as the truncation parameter k2 is increased?

On the λ Hyperparameter: The entropy regularization parameter λ seems crucial. The ablation shows a notable drop when λ is set to 10. How was the value of λ=100 chosen? Was this tuned on a validation set, and how sensitive is the model's performance to this value (e.g., for λ=50, λ=200)?

---

> ### Author Response · Authors · 2025-11-21
> **Official Comment by Authors for Reviewer wx4k (1/3)**
>
> We appreciate the reviewer’s detailed and constructive feedback. Below, we provide point-by-point responses to each concern. We have also revised the manuscript in accordance with your suggestions as well as comments from the other reviewers and uploaded an updated version. All changes made during revision are highlighted in blue for clarity.
> ***
> > **Q1. [Cost matrix C]** (Weaknesses 1) The entire semantic-awareness of the method hinges on the cost matrix C, which is built from the token embeddings of the reference SFT model. This is a reasonable choice, but its impact is not deeply explored. The ablation study only compares L2 vs. Cosine distance, but not the source or quality of the embeddings. The paper doesn't answer: what if the SFT model's embeddings are of poor quality? Would using embeddings from a more powerful, external model improve results?
>
> > (Questions 1) The semantic cost matrix C is fundamental to the method's success and is derived from the SFT model's embeddings. Have you experimented with using a different source for these embeddings, such as from a larger, more capable, and static model (e.g., GPT-4 embeddings)? Is it possible that a "better" or more semantically-grounded cost matrix C would lead to even larger gains?
>
> As the reviewer correctly noted, the token embeddings play a crucial role in defining the cost matrix $C$, which governs the semantic structure exploited by our Wasserstein regularizer. Because this cost must be computed over the vocabulary of the policy model, the embedding space could be aligned with the tokenizer of the target model; thus, only models sharing the same tokenizer can be used in a straightforward manner. For models that do not share a tokenizer (e.g., GPT-4 embeddings), obtaining token-level embedding alignment requires constructing a cross-token mapping, which is an interesting but nontrivial direction for future work.
>
> To examine the effect of embedding quality on WPR, we conduct an additional experiment using Gemma-2B and Gemma-7B models, which share the same tokenizer. For each model, we use the frozen token embeddings obtained after SFT to construct the cost matrix $C$, and independently varied both the policy backbone and the embedding source.
>
> [**Table 12 in the revised manuscript**: Win rates on TL;DR using the Gemma-based models, varying the policy backbone and the embedding spaces used to form the cost matrix]
> | Method                       | Backbone  | Embedding | Win rate (vs. SFT-2B) | Win rate (vs. RKL-2B) |
> |-----------------------------|-----------|-----------|------------------------|------------------------|
> | RKL-regularized PPO     | Gemma-2B  | -         | 0.848                  | -                      |
> |                             | Gemma-7B  | -         | **0.948**              | 0.668                  |
> | **Wasserstein-regularized PPO** | Gemma-2B | Gemma-2B | 0.924                  | 0.608                  |
> |                             | Gemma-2B  | Gemma-7B  | 0.908                  | 0.556                  |
> |                             | Gemma-7B  | Gemma-2B  | 0.944                  | 0.684                  |
> |                             | Gemma-7B  | Gemma-7B  | **0.948**              | **0.712**              |
>
> The results show that, across all configurations, WPR consistently outperforms RKL-regularized PPO when using the same policy backbone, indicating that WPR provides benefits regardless of the embedding model used. The size of the policy backbone has the largest effect on the performance, with Gemma-7B outperforming Gemma-2B.
>
> Interestingly, using the Gemma-7B embedding space to construct the cost matrix for the Gemma-2B policy does not surpass the performance achieved when using the original 2B embeddings. We conjecture that this occurs because the policy model is naturally grounded in the token geometry encoded by its own SFT embedding space, therefore this embedding space is naturally compatible with that model.
>
> We include this analysis and the additional results in Appendix D.1 in the revised manuscript.

---

> ### Author Response · Authors · 2025-11-21
> **Official Comment by Authors for Reviewer wx4k (2/3)**
>
> > **Q2. [Computational overhead]** (Weaknesses 2) The paper claims the overhead is low ("only 2.5%" increase in training time). However, this is after applying two aggressive truncations: k1=512 for the cost matrix and k2=128 for the policy distributions. This O(k_2^2) complexity is still per-token, per-step, compared to the O(1) cost of the KL penalty. A more detailed wall-clock time comparison and an analysis of the performance/compute trade-off (e.g., how much better does it get if k2=256?) would be beneficial.
>
> > (Questions 2) Could you please provide a more detailed breakdown of the computational overhead? The "2.5%" figure is encouraging, but a wall-clock time comparison per 1000 training steps (or per-epoch) against the RKL baseline would be very informative. How much does this overhead scale as the truncation parameter k2 is increased?
>
> As the reviewer notes, the KL penalty incurs only O(1) cost because the penalty depends solely on the probability of the sampled token, while the Wasserstein penalty incorporates semantic structure across tokens, resulting in higher complexity. However, in practice this additional cost is marginal compared to the forward and backward passes of a billion-parameter LLM. To quantify this, we report a detailed breakdown of the wall-clock time per 1,000 training steps, decomposed into (1) generation, (2) penalty computation, and (3) backpropagation. These measurements are obtained on 4 A100 GPUs, using the Gemma-2B policy model on the TL;DR dataset, with 8 batches per GPU and 8 gradient accumulation steps. As shown in the below table, the Wasserstein penalty computation is slower than KL (whose cost is nearly negligible at 0.005 hours), but its contribution remains small relative to the total training time.
>
> Note that while the time required to compute the regularization penalty differs across methods, the generation and training steps are independent of the choice of regularizer. Once the token-wise rewards, including the regularization penalty, are obtained, the forward generation and PPO updates proceed identically across regularization methods. In practice, the runtime of these components varies far more with the generated response length, and therefore we report unified timings for these stages.
>
> [**Table 15 in the revised manuscript**: Detailed breakdown of the wall-clock time (hours) per 1,000 training steps. The time required to compute the regularization penalty differs across methods, whereas the generation and training steps are independent of the regularization method and therefore reported using unified timings.]
> | | RKL    | WPR    |
> |--------------------|--------|--------|
> | Generation         | 0.769  | 0.769  |
> | Penalty computation| 0.005  | 0.117  |
> | Backpropagation    | 3.707  | 3.707  |
> | Total          | 4.481 | 4.593 |
> The analysis is included in Appendix D.3 in the revised version.
>
> We also conduct additional experiments to analyze how the overhead scales as the truncation parameter $k_2$ increases. As suggested, we vary $k_2$ and measure both the model performance and the penalty computation time. The results in the below table show that increasing $k_2$ from 64 to 128 improves performance with a predictable and moderate increase in computation time, when compared to the overall training time of about 4.5 hours per 1k steps. When increasing $k_2$ further from 128 to 256, performance slightly decreasing, suggesting that $k_2=128$ captures most of the probability mass in the token distribution and provides a sufficiently accurate approximation. For this reason, we use $k_2=128$ in all experiments.
>
> [**Table 13 in the revised manuscript**: Sensitivity analysis of the truncation hyperparameter $k_2$ on TL;DR with Gemma-2B. Time is the wall-clock time for the penalty computation.]
> | $k_2$  | Time (hours/1k steps) | Win rate (vs. SFT) | Win rate (vs. RKL) |
> |-----|------------------------|---------------------|---------------------|
> | 64  | 0.08                   | 0.864               | 0.528               |
> | 128 | 0.12                   | 0.924               | 0.608               |
> | 256 | 0.19                   | 0.916               | 0.584               |
>
> In the revision, we have added this analysis and the corresponding results to Appendix D.2.

---

> ### Author Response · Authors · 2025-11-21
> **Official Comment by Authors for Reviewer wx4k (3/3)**
>
> > **Q3. [Hyperparameter $\lambda$]** (Weaknesses 3) The method introduces a new, important hyperparameter λ, the entropy regularization strength, which is set to 100. The ablation study shows a performance drop when decreasing it to 10. This parameter's tuning and sensitivity are critical to the method's success, and a more in-depth discussion of its impact and how to set it would improve the paper's practical utility.
>
> > (Questions 3) The entropy regularization parameter λ seems crucial. The ablation shows a notable drop when λ is set to 10. How was the value of λ=100 chosen? Was this tuned on a validation set, and how sensitive is the model's performance to this value (e.g., for λ=50, λ=200)?
>
> As noted, $\lambda$ controls the amount of the entropic smoothing in the Sinkhorn distance and therefore determines that balance between faithfulness to the underlying cost matrix and the smoothness of the transport plan. A smaller $\lambda$ increases the contribution of the entropy term, producing overly soft transport plans and diminishing the effect of the semantic structure encoded in $C$. This behavior explains the degradation observed when reducing $\lambda$ from 100 to 10 in Table 3 of the original main manuscript. Conversely, excessively large values of $\lambda$ cause the kernel $K=\exp (-\lambda C)$ to approach a near-zero matrix, leading to numeric oscillations during the Sinkhorn row/column updates. Thus, $\lambda$ could be chosen to balance semantic fidelity and numerical stability.
>
> To the best of our knowledge, our work is the first to apply an entropy-regularized Wasserstein penalty to the LLM token space in the RL fine-tuning stage. Therefore, we initially explored several values empirically and found that $\lambda=100$ provided stable behavior during training. This value is used for all experiments in the manuscript, where it consistently produced superior performance across tasks.
> We additionally provide a sensitivity analysis with $\lambda= \{ 50, 100, 200 \}$. As shown in the below table, WPR achieves a win rate (vs. RKL) of at least 0.5 for all tested values, demonstrating that it consistently outperforms RKL-based regularization. As expected, smaller $\lambda$ leads to a performance drop due to reduced influence of semantic information, consistent with the above analysis.
>
> [**Table 14 in the revised manuscript**: Sensitivity analysis of the entropy regularization parameter $\lambda$ on TL;DR with Gemma-2B.]
> | $\lambda$   | Win rate (vs. SFT) | Win rate (vs. RKL) |
> |-----|---------------------|---------------------|
> | 50  | 0.900               | 0.564               |
> | 100 | 0.924               | 0.608               |
> | 200 | 0.916               | 0.612               |
>
> Appendix D.2 in the revised manuscript contains this analysis and the corresponding additional results.

---

### Author Response · Authors · 2025-12-03
**Author Final Remarks by Authors**

We would like to thank the reviewers for their thoughtful comments and insightful suggestions. We greatly appreciate the consistent recognition of the strengths of our work, including its **natural and novel motivation** (wx4k, rU8j, AZq3), **rigorous theoretical formulation** (wx4k, Ndnt, AZq3), **consistent empirical improvements** (wx4k, rU8j, AZq3), and **comprehensive analyses** (wx4k, Ndnt).


Below, we summarize the main concerns raised in the reviews and the corresponding results.

***

* **Semantic awareness analysis** (rU8j Q1, Ndnt Q2, AZq3 Q1&Q3)
  * **[Quantitative validation of semantic relationships]** To empirically verify that WPR reflects semantic similarity, we use BERTScore as a quantitative measure of semantic equivalence. As shown in **Table 7** of the revised manuscript, the Wasserstein penalty exhibits a stronger correlation with BERTScore than the KL penalty. This provides empirical evidence that WPR better respects semantic similarity.
  * **[Qualitative and quantitative analysis of model behavior]** To clarify how semantic structure affects model behavior:
    * **Case study** (**Figure 6** in the revised manuscript): WPR yields low penalties for meaning-preserving shifts and high penalties for semantically unrelated shifts.
    * **Semantic coherence of top-k predictions** (**Table 8** in the revised manuscript): WPR produces next-token candidate sets with higher semantic coherence than KL-based regularization.

    * These results suggest that WPR penalizes semantic drift and encourages coherent semantic structure in the token prediction distribution. This likely helps prevent unnecessary changes when the learned policy is already semantically close to the reference policy. It allows the model to maintain meaning-preserving behavior while still permitting reward-driven shifts when necessary.

***

* **Generalization and scalability** (rU8j Q3, Ndnt Q1, AZq3 Q2)

We expanded our experiments to include:

1. **Larger models**: Gemma-7B on the summarization task (**Table 2** in the revised manuscript).
2. **Different LLM families**: Qwen-1.5-1.8B-Chat on the dialogue generation task (**Table 3** in the revised manuscript).
3. **Different tasks**: CodeGemma-7B on the code generation task (**Table 5** in the revised manuscript).

These results demonstrate that WPR consistently outperforms KL-based regularization across architectures, scales, and tasks. This highlights the robustness and broad applicability of WPR.

***

* **Analysis of embedding space for cost matrix** (wx4k Q1, rU8j Q2)

We added a new experiment to **Table 12** in the revised manuscript, analyzing how the source of token embeddings impacts the effectiveness of WPR. Across all configurations, WPR improves over KL-based regularization under the same backbone. Larger backbones naturally perform better; however, interestingly, using Gemma-7B embeddings for a Gemma-2B policy model does not outperform using the policy’s own embeddings. We conjecture that the policy model is naturally grounded in the token geometry encoded by its own SFT embedding space.

***

We sincerely thank all reviewers again for their constructive comments. We hope that the additional explanations and analyses address the concerns.

---

### Meta-Review · Area_Chair_vMXd · 2026-01-05

**Summary:**

While the paper is technically sound and empirically strong, the divergence in reviewer assessments appears to stem from the absence of a clearly articulated conceptual hypothesis linking the proposed Wasserstein regularization to semantic behavior. Although the derivations are correct and the method is well engineered, reviewers consistently struggle to evaluate what exactly is being claimed beyond performance gains—whether the token-space geometry is meant to approximate semantics, encode model-specific inductive bias, or merely act as a smoother proxy regularizer. In the absence of such a hypothesis, concerns about the cost matrix, embedding dependence, semantic validation, and generality surface in different forms across reviews, and the rebuttal largely addresses these in isolation rather than resolving the underlying ambiguity. Despite this issue, the conceptual contribution and interesting experimental results do justify acceptance.

**Reviewer Concerns:**

Across reviews, the primary source of disagreement appears to be not technical soundness but the absence of a clear conceptual hypothesis linking the proposed Wasserstein regularization to semantic behavior. While the authors respond constructively by adding broader experiments, detailed runtime and memory analyses, and several semantic diagnostics (e.g., BERTScore correlations, embedding-coherence metrics, and case studies), these additions remain indirect and partly circular, as they rely on the same tokenizer- and embedding-specific geometry used to define the cost matrix. Reviewers consistently note that this cost matrix remains a black box whose semantic validity, failure modes, and generality across tokenizers or embedding spaces are not characterized, even though the rebuttal improves transparency and documentation. As a result, despite solid derivations and improved empirical coverage, the paper does not fully resolve whether the observed gains arise from genuine semantic awareness or from model-specific regularization effects, which likely explains the split reviewer reception.

**Reviewer Scores:**

See above.

---

### Decision · Program_Chairs · 2026-01-26

Accept (Poster)